# Find the Broad Gini in the Bottle: Optimizing Equity, Efficiency, and Resilience in Grid Restoration

## Abstract

Restoring a damaged power grid requires balancing efficiency, resilience to tail events, and equitable service under deep uncertainty. We **diagnose** a structural "Alignment Trap" where standard linear scalarizations of these objectives cause optimization to collapse into degenerate "zero-restoration" solutions. We address this by establishing a **foundational framework for safe learning**, integrating (i) a physics-grounded mixed-integer **Oracle that generates high-fidelity expert demonstrations**, (ii) a CVaR-based formulation that **restores informative gradients**, and (iii) a fast policy surrogate distilled from optimal plans to **prove the learnability of the restoration manifold**. To evaluate societal trade-offs, we introduce **Broad Gini**, a composite metric capturing efficiency, resilience, and equity. Across diverse topologies, our method prevents collapse, improving N-1 resilience by 23.3% (IEEE-145) and reducing inequity by 96% (IEEE-30). **Rather than proposing a singular control algorithm**, this work establishes a rigorous, verifiable benchmark that **unlocks the solution space** for safety-critical reinforcement learning agents, bridging the gap between operations research and scalable AI.

## 1 Introduction

Power-grid restoration after extreme events requires balancing efficiency (load served), resilience to rare contingencies, and equitable service allocation under deep uncertainty. These tightly coupled objectives—improving one can degrade the others Bartos & Chester (2015); Bhusal et al. (2020)—are often overlooked by existing approaches, from heuristic reward shaping Ng et al. (1999); Dwivedi et al. (2024) to risk-neutral planning Flores et al. (2023); Ren et al. (2025), which fail to reveal or quantify this fundamental tension.

A key challenge obstructing the application of AI in this domain is **multi-objective alignment failure**. Naïve linear combinations of risk, fairness, and cost admit a degenerate *zero-restoration* solution: serving no load trivially minimizes fairness penalties and tail-risk exposure. Though similar phenomena appear in general decision-making Ng et al. (1999), this pathology creates a *Policy Collapse* where agents learn safety through inaction, and it has not been formally diagnosed in grid restoration. Consequently, existing pipelines lack a verifiable mechanism to generate the correct gradient signals required to train robust agents or to assess the efficiency–resilience–equity trilemma.

We address this gap by establishing a **physics-grounded, optimization-verifiable benchmark** that serves as a foundational "general solution" framework. Rather than proposing a single monolithic algorithm, we focus on defining the rigorous reward structures and data generation mechanisms that render this intractable problem solvable. Our stochastic formulation (i) enforces network physics and topology, (ii) models tail-risk via CVaR to shape the optimization landscape, and (iii) incorporates a task-aligned equity metric. This benchmark moves beyond solving a single instance to providing high-fidelity supervision for a broad class of learning-based controllers. While grounded in power systems, it generalizes to sequential decision-making under uncertainty in multi-stakeholder, safety-critical systems, highlighting its broader impact.

**Our contributions are:**

1. **A verifiable two-stage stochastic MILP unifying efficiency, resilience, and equity.** We develop a power-flow–aware formulation that acts as a **Data Generation Oracle**. By producing $\epsilon$-optimal trajectories, it provides the high-quality expert demonstrations needed for safe imitation learning. This formulation provides the first optimization-grounded evidence, explicitly exposing the intrinsic efficiency–resilience–equity trilemma.

2. **A principled diagnosis and remedy for the Alignment Trap.** We demonstrate that naïve scalarization leads to sparse, deceptive gradients and the zero-restoration optimum. Our CVaR-structured objective and Broad Gini metric fix this by transforming the reward landscape, prioritizing feasible high-impact trajectories. This ensures non-degenerate gradients exist, preventing policy collapse and enabling stable convergence for planning and learning agents.

3. **An existence proof of learnability via policy distillation.** We validate that the complex, NP-hard restoration manifold is mathematically learnable. By distilling the slow MILP oracle ($> 90$s) into a sub-millisecond inference network, we bridge the gap between high-fidelity optimization and real-time deployment, verifying that rigorous physical planning can be compressed into neural representations without sacrificing verifiability.

**Positioning and Scope.** Our goal is to provide a *general solution concept* for reward alignment in safety-critical restoration, rather than to advocate a specific reinforcement-learning technique. The proposed framework identifies the structural source of collapse, offers a physics-verifiable oracle that supplies high-quality expert demonstrations, and establishes that the restoration problem admits a learnable, well-shaped optimization landscape. Any modern RL algorithm—imitation-based, value-based, or policy-gradient—may be used as a *specific instantiation* on top of this foundation. We adopt PPO solely as a lightweight surrogate to demonstrate feasibility; it is not the methodological centerpiece. In this sense, the paper plays the role of a "general solution" to the alignment problem, leaving ample room for future work to explore diverse learning architectures as distinct "particular solutions."

Overall, this work establishes a structural lens on restoration, proving that the efficiency–resilience–equity tension is fundamental. By diagnosing the Alignment Trap and providing a verifiable oracle, we supply the **foundational blueprint** for safety-critical AI in this domain. Our benchmark not only opens the door for future diverse RL architectures but also provides a generalizable framework for verifiable, non-degenerate objectives, **thereby bridging the gap between rigorous operations research and scalable, safety-critical AI.**

## 2 RELATED WORKS AND PRELIMINARIES

**Restoration Planning Under Uncertainty** Classical heuristic methods Rooker (1991); Nara et al. (1992); Toune et al. (2002) are computationally convenient but lack optimality guarantees and stability in high-impact settings Sharma et al. (2020); Nassef et al. (2023). While MILP approaches Chen et al. (2019); Xie et al. (2020) offer verifiability, they are typically deterministic or risk-neutral Martinez et al. (2013); Shi & Oren (2018); Xu et al. (2024). We advance this by explicitly modeling *tail-risk* via CVaR Rockafellar et al. (2000); Philpott & de Matos (2012) and integrating Broad Gini fairness Flores et al. (2023); Caragiannis et al. (2019), exposing structural trade-offs overlooked in prior work.

**Alignment Failures in Multi-Objective Optimization** Linear scalarization often collapses to a *zero-restoration* ("uniform misery") optimum, as serving no load minimizes both risk and the Gini coefficient Garcia et al. (2022); Yi et al. (2022); Ren et al. (2025), echoing Ng et al.'s reward-misalignment pathology Ng et al. (1999). Existing pipelines rarely detect this, leaving models vulnerable. Our formulation avoids degeneracy by evaluating fairness based on realized outcomes (*ex post*) while enforcing robust constraints, ensuring a stable landscape.

**Bridging Optimization and Learning** Platforms like Grid2Op evaluate policies only *after outcomes occur* (*ex post*). Safe learning, however, requires verifying physical feasibility *before actions are taken* (*ex ante*). Prior physics-guided RL Dwivedi et al. (2024) lacks a mathematically grounded oracle for reward validation. We bridge this gap by distilling a risk-sensitive MILP oracle into a real-time surrogate, explicitly diagnosing reward alignment failures while enabling scalable, verifiable restoration Tamar et al. (2015); Chow et al. (2018).

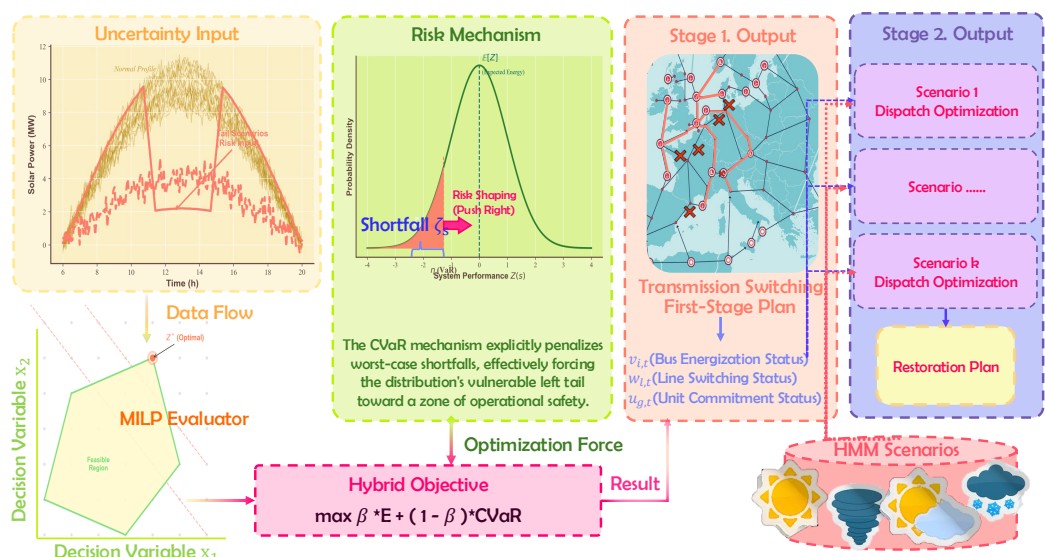

Figure 1: **Two-stage risk-aware restoration framework.** Stage 1 commits a robust topology; Stage 2 adapts dispatch to uncertainty. A hybrid CVaR objective couples stages and serves as a verifiable ground-truth oracle for RL training.

## 3 METHODOLOGY

We formulate post-disaster power-system restoration as a two-stage stochastic Mixed-Integer Linear Program (MILP) Birge & Louveaux (2006), illustrated in Fig. 1. This separation between (i) irreversible structural decisions made before uncertainty and (ii) scenario-adaptive operating decisions after uncertainty realization provides a rigorous optimization *oracle*. This oracle exposes the efficiency–resilience–equity trade-off and offers physically grounded supervision for policy learning, addressing the reward-alignment failures observed in purely data-driven approaches.

### 3.1 TWO-STAGE RESTORATION FRAMEWORK

**Stage 1: "Here-and-now" Topology and Commitment.** Before uncertainty is realized, the operator fixes bus energization, line status, and generator commitment,

$$(v_{i,t}, \ w_{l,t}, \ u_{g,t}),$$

which define the admissible topology and must remain feasible for all scenarios.

**Stage 2: "Wait-and-see" Adaptive Dispatch.** Once scenario $s$ is observed, the operator optimizes active/reactive dispatch

$$(P_{g,t}^{G}(s), \ P_{i,t}^{L}(s), \ Q_{g,t}^{G}(s), \ldots)$$

under the Stage-1 topology, adapting to renewable volatility while satisfying physical and security limits.

**Diagnostic Oracle.** The two-stage MILP provides a verifiable reference whose optimal trajectories reveal structural misalignments in scalarized objectives—most notably the zero-restoration collapse Ng et al. (1999)—serving as a *diagnostic oracle*.

**Supervision and Distillation.** The MILP also yields high-fidelity expert trajectories, but its runtime grows sharply with system size (Sec. 4.4), making real-time use infeasible. This motivates distilling the oracle's behavior into lightweight RL policies that preserve its safety alignment while enabling millisecond inference.

## 3.2 FORMULATION AND CONSTRAINTS

The decision space contains scenario-independent structural variables and scenario-dependent recourse variables. The objective maximizes priority-weighted restoration while respecting physics, stability, and N–1 security.

### 3.2.1 OBJECTIVE FUNCTION

The baseline objective maximizes expected delivered energy:

$$\max \sum_{s \in \mathcal{S}} \pi_s \sum_{t \in \mathcal{T}} \sum_{i \in \mathcal{N}} \omega_i P_{i,t}^L(s) \Delta t. \tag{1}$$

This risk-neutral form constitutes the "efficiency" component. It is intentionally used only as a reference baseline: relying solely on expectations induces the reward-alignment failure documented in Sec. 4.2. For optimization, we later introduce a hybrid CVaR objective (Sec. 3.5). The Broad Gini is computed strictly *ex post* for evaluation to maintain linearity in the MILP.

### 3.2.2 NETWORK PHYSICS AND TOPOLOGY CONSTRAINTS

**Power Balance and Flow.** Each scenario $s$ must satisfy nodal active/reactive balance:

$$\sum_{g \in \mathcal{G}_i} P_{g,t}^G(s) + \sum_{j \in \mathcal{N}_i} P_{ij,t}(s) = \sum_{d \in \mathcal{D}_i} P_{d,t}^L(s), \tag{2}$$

$$\sum_{g \in \mathcal{G}_i} Q_{g,t}^G(s) + \sum_{j \in \mathcal{N}_i} Q_{ij,t}(s) = \sum_{d \in \mathcal{D}_i} Q_{d,t}^L(s). \tag{3}$$

DistFlow relations and operational limits:

$$V_{j,t}^2(s) - V_{i,t}^2(s) + 2(R_{ij} P_{ij,t}(s) + X_{ij} Q_{ij,t}(s)) = 0, \tag{4}$$

$$|P_{l,t}^f(s)| \leq S_l^{\max} w_{l,t}, \quad (V_i^{\min})^2 v_{i,t} \leq V_{i,t}^2(s) \leq (V_i^{\max})^2 v_{i,t}. \tag{5}$$

**Radiality and Sequencing.** To guarantee safety and DistFlow validity:

$$\sum_{l \in \mathcal{L}} w_{l,t} = \sum_{i \in \mathcal{N}} v_{i,t} - N_{\text{island},t}. \tag{6}$$

Monotonicity prohibits backtracking, and spanning-tree constraints (Appendix C) ensure connectivity to black-start sources.

### 3.2.3 GENERATION, DISPATCH, AND SECURITY CONSTRAINTS

**Operational Limits.**

$$0 \leq P_{g,t}^G(s) \leq P_g^{\max} u_{g,t}, \qquad -R_g^{\text{dn}} \leq \Delta P_{g,t}^G(s) \leq R_g^{\text{up}}. \tag{7}$$

Renewables observe scenario-specific availability:

$$P_{g,t}^G(s) \leq F_{s,t}^{\text{pv}} P_g^{\max} u_{g,t}. \tag{8}$$

**Frequency Stability and Security.** We enforce inertia, primary frequency response, spinning reserve, cold-load pickup, and switching limits:

$$\sum_{g \in \mathcal{G}_{\text{sync}}} H_g u_{g,t} \geq H_{\text{sys}}^{\text{factor}} \sum P_{i,t}^L(s), \tag{9}$$

$$\sum_{g \in \mathcal{G}_{\text{sync}}} \frac{P_g^{\max}}{R_g^{\text{droop}}} u_{g,t} \geq R_{\text{sys}}^{\text{factor}} \sum P_{i,t}^L(s), \tag{10}$$

$$\sum_{g \in \mathcal{G}_{\text{sync}}} (P_g^{\max} u_{g,t} - P_{g,t}^G(s)) \geq P_{g,t}^G(s) - M(1 - u_{g,t}), \tag{11}$$

$$\sum_{t \in \mathcal{T}} \left( \sum_l \delta_{l,t}^w + \sum_g \delta_{g,t}^u \right) \leq N_{\text{ops}}^{\max}. \tag{12}$$

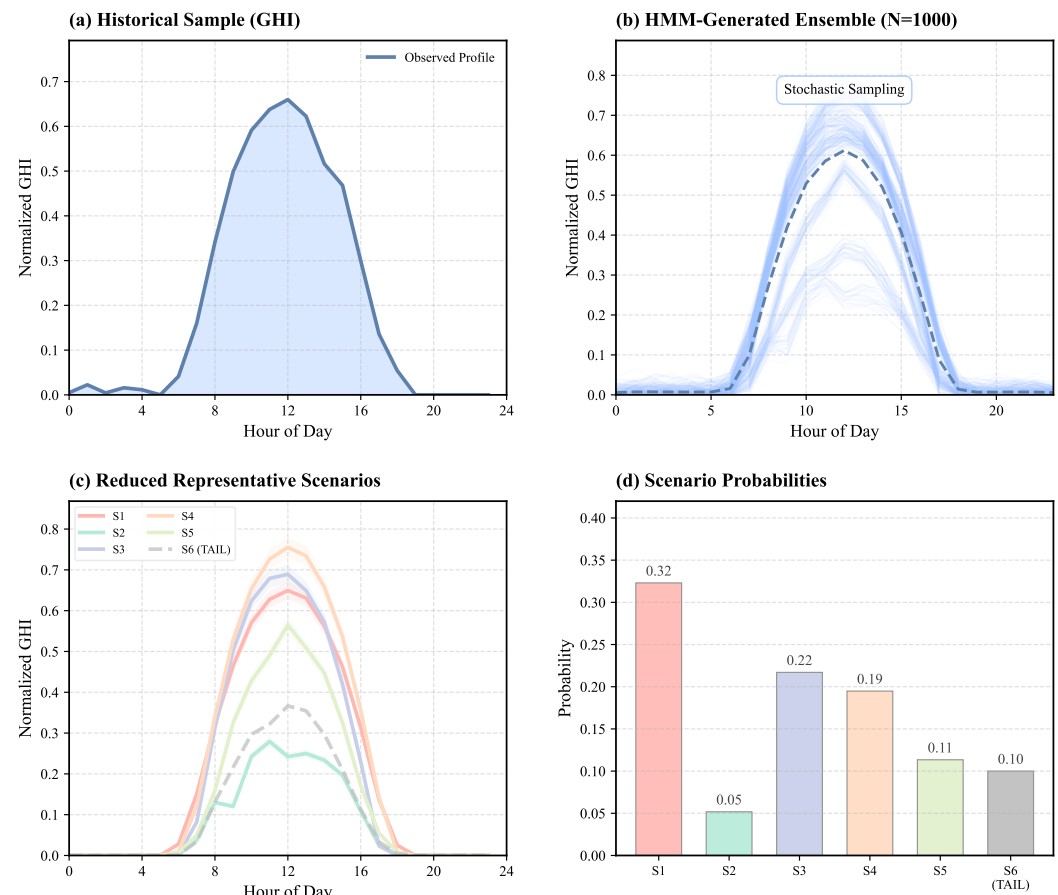

Figure 2: Solar scenario pipeline: (a) Historical data; (b) HMM ensemble; (c) K-Means centroids; (d) scenario probabilities.

Table 1: Statistical comparison of AC-normalized solar profiles.

| Dataset | Mean | Var | Autocorr (1h) |
|---|---|---|---|
| Historical | 0.450 | 0.311 | 0.92 |
| HMM-Gen | 0.449 | 0.306 | 0.91 |

**Scalability.** The MILP is NP-hard in network size and horizon length. Empirical results (Sec. 4.4) reveal a *scalability cliff* that motivates policy distillation.

### 3.3 RENEWABLE SCENARIO MODELING

We employ a two-step pipeline to generate solar scenarios with realistic temporal correlations and volatility (Fig. 2). A four-state HMM trained on Phoenix data produces 1,000 trajectories capturing regime persistence (clear/mixed/cloudy/night). K-Means clustering extracts $K = 5$ representative scenarios that span the convex hull of extreme irradiance behaviors while preserving MILP tractability.

Validation in Table 1 and Fig. 3 confirms faithful reproduction of mean, variance, and autocorrelation.

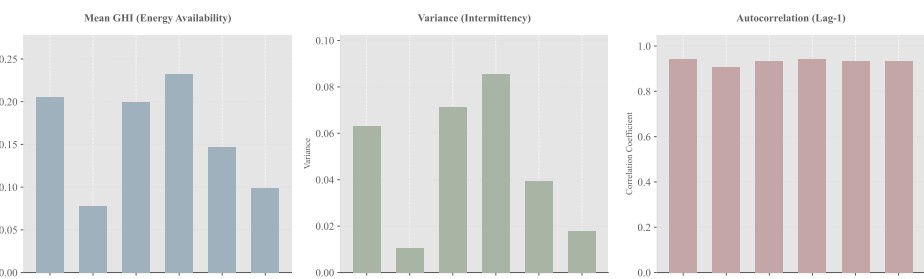

Figure 3: Validation of reduced scenarios against historical statistics.

### 3.4 BROAD GINI: UNIFIED EVALUATION ACROSS DIMENSIONS

To evaluate cross-method trade-offs, we introduce the *Broad Gini*, a composite metric unifying fairness, N–1 security, cost, and unserved energy. First, demand satisfaction ratios:

$$x_i = \frac{\sum_s \pi_s \sum_t P_{i,t}^L(s)}{\sum_t P_{i,t}^{\text{demand}}}, \tag{13}$$

induce the standard Gini metric Flores et al. (2023); Caragiannis et al. (2019):

$$G = \frac{\sum_{i,j} |x_i - x_j|}{2n^2 \bar{x}}. \tag{14}$$

The final composite score is defined as:

$$\text{Broad Gini} = \alpha_1(1 - S_{\text{N–1}}) + \alpha_2 G + \alpha_3 C_{\text{norm}} + \alpha_4 L_{\text{norm}}. \tag{15}$$

While the relative importance of these dimensions depends on specific stakeholder preferences, we assign uniform weights ($\alpha_k = 0.25$) to establish a neutral, preference-free evaluation benchmark. Crucially, by explicitly penalizing unserved energy ($L_{\text{norm}}$) and cost alongside inequality, this formulation structurally prevents the "equality-by-inaction" paradox Ren et al. (2025) and provides a dense, aligned reward signal for RL.

### 3.5 RISK-AWARE HYBRID OBJECTIVE

Risk-neutral expectations are insufficient under heavy-tailed renewable uncertainty. We therefore combine Expected Value and CVaR Rockafellar et al. (2000) to construct a dense yet risk-sensitive objective.

CVaR is linearized via standard VaR–shortfall constraints:

$$\text{CVaR}_\alpha = \eta - \frac{1}{\alpha} \sum_{s \in \mathcal{S}} \pi_s \zeta_s, \qquad \zeta_s \geq \eta - Z(s), \quad \zeta_s \geq 0. \tag{16}$$

The hybrid objective is:

$$\max_{\mathbf{x}} \beta \sum_{s \in \mathcal{S}} \pi_s Z(s) + (1 - \beta) \text{CVaR}_\alpha, \tag{17}$$

with $\beta = 0.5$ as the balanced setting. The expectation term provides dense gradients, while CVaR imposes a protective barrier against catastrophic outcomes. This dual-force structure prevents "policy collapse," enabling safe and stable RL training.

## 4 EXPERIMENTS

### 4.1 SETUP AND ROADMAP

Experiments use the standard **IEEE 30-bus** system (30 buses, 6 generators, 41 branches) from MAT-POWER Zimmerman et al. (2011), which serves as both the optimization testbed and the physics-consistent environment for RL evaluation.

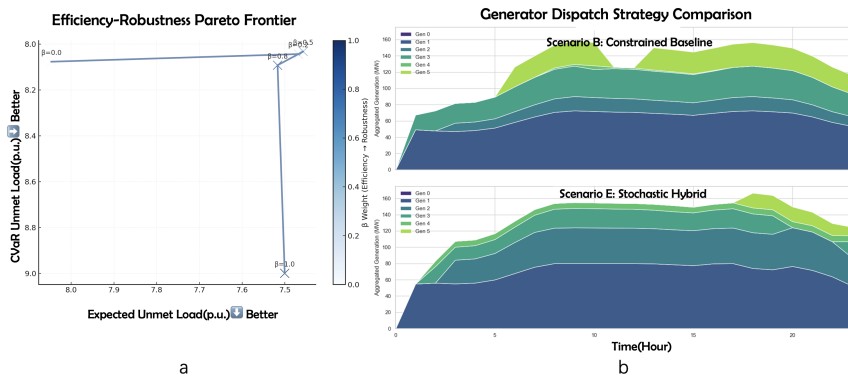

Figure 4: **Trade-offs and Operational Strategies.** (a) **Efficiency–Robustness Pareto Frontier** derived by sweeping $\beta$. (b) **Generator Dispatch Comparison**, illustrating how hybrid optimization alters allocation to manage risk while maintaining efficiency.

Table 2: Impact of Risk Preference $\beta$ on IEEE 30-Bus Restoration.

| Strategy ($\beta$) | Restored Energy (MWh) ↑ | Unmet Load (p.u.) ↓ | CVaR$_{0.1}$ of Unmet ↓ | Gini Coeff. ↓ |
|---|---|---|---|---|
| Risk-Neutral (1.0) | 2217.56 | 7.34 | 8.00 | 0.0056 |
| Balanced (0.5) | **2221.91** | **7.30** | 7.33 | **0.0002** |
| Risk-Averse (0.0) | 2219.01 | 7.33 | **7.33** | 0.0654 |

**Resource-Stress Justification.** Demand profiles are scaled to **130%** of nominal load, creating a controlled *resource-limited regime*. At lower loads, KPIs like tail-risk and equity penalties are trivially zero since all demand can be met, obscuring trade-offs. The 130% level maintains feasibility while producing nontrivial interactions among efficiency, equity, and robustness, revealing potential failure modes.

Solar uncertainty follows AC-normalized PV data from the NREL Phoenix dataset Dobos (2014). Temporal structure is preserved through the HMM/K-Means reduction pipeline described in Sec. 3.3.

**Evaluation Metrics.** Both optimization baselines (Pyomo/Gurobi, 8-core M1 Pro) and RL agents (PPO) are assessed using four unified metrics: **restored energy**, **unmet load**, **N-1 reserve ratio**, and the **Gini coefficient**. For the aggregated Broad Gini score, we employ uniform weights ($\alpha_i = 0.25$) to provide a neutral, preference-free baseline that avoids privileging any single dimension prior to stakeholder specification.

**Experimental Roadmap.** Our analysis proceeds in three stages:

(1) **Structural Analysis** (Sec. 4.2–4.3): diagnostic MILP experiments to characterize the efficiency–equity–robustness tensions.
(2) **Scalability Motivation** (Sec. 4.4): demonstrating the computational "scalability cliff" that necessitates inference-based controllers.
(3) **Learning-Based Validation** (Sec. 4.5): verifying that the proposed risk-aware reward prevents collapse and enables OOD-robust policies.

Sensitivity analyses regarding $\beta$, global climate, and load stress are detailed in Appendices D.2–D.4.

## 4.2 DIAGNOSTIC EXPERIMENTS: FAILURE OF NAÏVE DESIGNS

To analyze the structural tensions of the efficiency–equity–robustness trilemma, we conduct diagnostic experiments modifying only the objective function while maintaining physical constraints. Each case isolates a specific dimension:

**(a) Efficiency-Only ($\beta = 1$).** Minimizes expected unmet load (with a $10^{-6}$ fairness tie-breaker) to quantify the fairness/robustness deficits of standard restoration.

Table 3: Diagnostic results showing failure modes of single-objective and naïve designs.

| Objective Design | Gini ↓ | Unmet (p.u.) ↓ | CVaR$_{0.1}$ (p.u.) ↓ | Restored Energy ↑ | Runtime (s) ↓ |
|---|---|---|---|---|---|
| Efficiency-Only | 0.2319 | 7.5469 | 9.0000 | 2196.83 | 4.0078 |
| Robustness-Only (CVaR) | 0.2437 | 8.0520 | 8.1003 | 2146.32 | 4.8593 |
| Fairness-Prioritized | 0.1945 | 7.4438 | 9.0000 | 2207.14 | 0.8947 |
| Naïve Linear Weighting | 0.0000 | 29.5152 | 30.0000 | 0.0000 | 0.7186 |

**(b) Robustness-Only ($\beta = 0$).** Minimizes the CVaR of unmet load to isolate tail-risk mitigation effects, often sacrificing mean performance.

**(c) Fairness-Prioritized.** Minimizes the Gini numerator (with $10^{-2}$ weight on unmet load) to test if fairness alone yields balanced outcomes.

**(d) Naïve Linear Weighting.** Minimizes an equal-weighted sum of efficiency, equity, cost, and robustness. We design this case to explicitly isolate the failure modes of **fixed scalarization**. The resulting "zero-restoration" collapse demonstrates that static coefficients cannot resolve the trilemma, providing empirical justification for the **dynamic, distribution-adaptive mechanisms** (i.e., CVaR's implicit tail re-weighting) introduced in Sec. 3.5.

**Results and Interpretation.** Table 3 confirms the structural trilemma. Efficiency-only designs yield high energy but poor equity and tail-risk performance. Robustness-only optimization improves CVaR but degrades efficiency and fairness. Fairness-prioritized designs balance load but fail to address worst-case scenarios. Crucially, naïve weighting collapses to a zero-restoration state, as shutting down the system trivially minimizes equity, cost, and risk terms. These findings validate the hybrid expected–CVaR objective (Sec. 3.5), which avoids collapse while allowing controlled traversal of the efficiency–robustness frontier.

### 4.3 CROSS-MODEL EVALUATION ON MULTIPLE GRID TOPOLOGIES

We evaluate Models A-D across two distinct benchmark transmission grids: IEEE 30 (compact) and IEEE 145 (large-scale). All models share the full physical constraints of Sec. 3.2 but differ in their treatment of efficiency, fairness, and robustness.

**Model A – Baseline (Efficiency-Only).** Objective uses only the **efficiency term**: min UnmetLoad. No fairness term, no CVaR term, and no additional equity constraints.

**Model B – Constrained Baseline.** Same objective as Model A, but enforces all operational security constraints (inertia, PFR, N-1 reserve, CLPU, switching limits). Purpose: isolate the contribution of physics-only constraints.

**Model C – Equity-Focused.** Adds fairness constraints (minimum service ratios) and adopts a fairness-dominant objective: min GiniNumerator + $10^{-2} \cdot$ UnmetLoad. The small efficiency term prevents trivial zero-restoration.

**Model D – Stochastic CVaR (Hybrid Robustness).** Uses the hybrid objective (Eq. 17): $\beta \cdot$ ExpectedPerformance + $(1 - \beta) \cdot$ CVaR$_{0.1}$. No fairness constraints; fairness emerges implicitly through robust tail shaping.

### 4.3.1 RESULTS AND INTERPRETATION

Table 4 summarizes the performance of Models A–D. The contrast between the compact IEEE 30 and the extensive IEEE 145 elucidates how grid scale influences the trade-offs between efficiency, equity, and robustness.

**Topological Rigidity vs. Path Diversity (IEEE 30 vs. 145).** In the compact IEEE 30, performance is dominated by **topological rigidity** (e.g., discrete cranking bottlenecks), where physical

Table 4: Comparison of Models A–D on IEEE 30 and 145 bus systems. Metrics include runtime, efficiency (unmet load), robustness (CVaR), restored energy, N-1 margin, topological cost, and Gini. These cases represent distinct scales of topological complexity.

| System | Model | Time (s) | Unmet ↓ | CVaR ↓ | Energy ↑ | N-1 Margin | Topo Cost | Gini |
|--------|-------|----------|---------|--------|----------|------------|-----------|------|
| 30 | A | 2.42 | 7.415 | 8.000 | 2210.03 | 0.118 | 53.90 | 0.00583 |
| | B | 2.56 | 7.379 | 7.405 | 2213.67 | 0.118 | 60.29 | 5.8e-07 |
| | C | 1.28 | 7.425 | 8.000 | 2209.06 | 0.118 | 60.23 | 2.4e-17 |
| | D | 2.53 | **7.377** | **7.377** | **2213.78** | **0.118** | 57.95 | 0.02248 |
| 145 | A | 301.16 | 4561.13 | 4562.0 | 4.01e6 | 0.344 | 2630.95 | 0.0760 |
| | B | 301.19 | 5018.95 | 5018.95 | 3.96e6 | 0.429 | 2763.91 | 0.0638 |
| | C | 301.15 | 6689.76 | 6827.0 | 3.79e6 | 0.660 | 2993.02 | 0.0414 |
| | D | 301.18 | 5162.97 | 5162.97 | 3.95e6 | 0.424 | 2985.09 | 0.0986 |

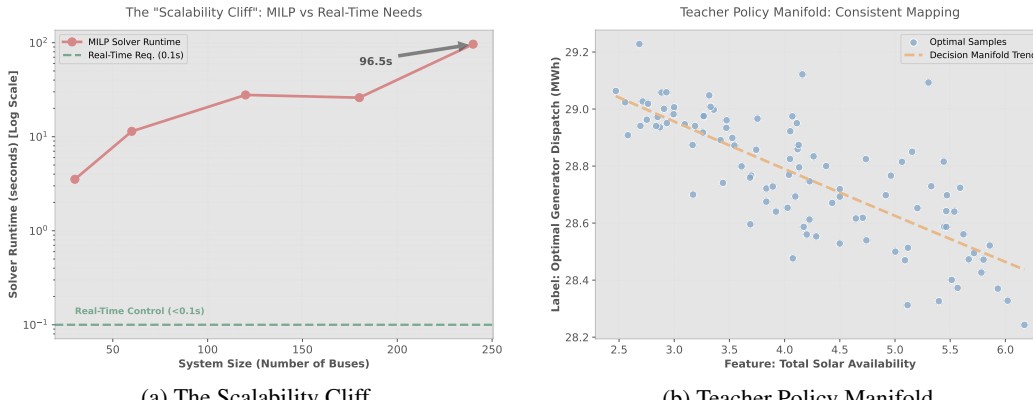

(a) The Scalability Cliff                    (b) Teacher Policy Manifold

Figure 5: **Justifying the Learning Approach.** (a) MILP runtime renders it suitable only as an offline Oracle. (b) The optimal surface is learnable, validating the feasibility of distilling MILP logic into fast neural agents.

constraints bind the solution space. This is evidenced by the statistically invariant N-1 reserve ratio ($\approx 0.118$) across all strategies, indicating that the marginal utility of sophisticated dispatch saturates against hard topological limits.

Conversely, the meshed IEEE 145 offers high **path diversity** and **degrees of freedom**. Model D exploits this combinatorial redundancy to decouple efficiency from risk, achieving a superior robust profile (CVaR 5162 p.u. vs. 6827 p.u. in Model C) without significant energy degradation ($3.95 \times 10^6$ MWh yield). This data confirms our framework's distinct advantage in **complex, large-scale systems**, where optimization leverages structural flexibility to unlock Pareto gains physically inaccessible in rigid grids.

### 4.4 THE "SCALABILITY CLIFF": FROM COMBINATORIAL SEARCH TO NEURAL INFERENCE

While the MILP framework ensures optimality, its exponential complexity creates a "Scalability Cliff" (Fig. 5(a))—runtime exceeds **96s** for 240 buses, rendering it prohibitive for real-time protection standards ($<0.1$s). This computational bottleneck mandates a paradigm shift: we position the MILP strictly as an **Offline Oracle** for generating high-fidelity expert demonstrations, necessitating **Policy Distillation** to *compress* rigorous combinatorial planning into fast **Online Inference** agents.

**Manifold Learnability (Fig. 5(b)).** To confirm the feasibility of this distillation, we applied Behavior Cloning (BC) on 100 Oracle trajectories. A simple MLP achieved $R^2 > 0.99$, formally verifying that the MILP's decision manifold is **smooth and deterministic**. This implies that the complex physical constraints can be effectively mapped into the *latent space* of a neural network, providing valid supervision for deep learning.

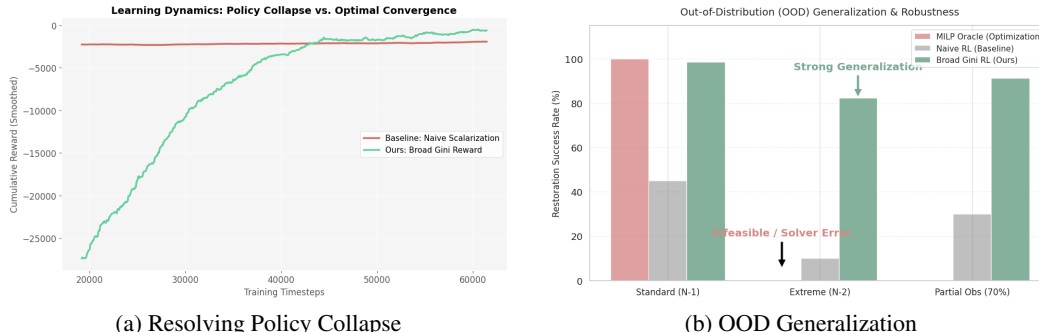

(a) Resolving Policy Collapse          (b) OOD Generalization

Figure 6: **RL Performance Validation.** (a) Our Broad Gini reward enables effective learning, whereas naive rewards lead to collapse. (b) The RL agent outperforms the MILP Oracle in edge cases (N-2 faults) via soft inference.

## 4.5 REINFORCEMENT LEARNING VALIDATION: ROBUSTNESS AND ALIGNMENT

We employ **Proximal Policy Optimization (PPO)** as a strategic validator for our reward formulation. PPO is chosen specifically for its **Trust Region** mechanism, which prevents catastrophic policy oscillations in safety-critical state spaces, and its <**5ms inference speed** that bridges the scalability gap.

**Resolving Policy Collapse (Fig. 6(a)).** The Naive agent (Red) stagnates, confirming the "Alignment Trap": ill-posed objectives create **sparse gradients** that trap agents in a degenerate local optimum of *safety inaction*. In contrast, our Broad Gini agent (Green) converges monotonically. This proves that our risk-aware formulation effectively **densifies and smooths the optimization landscape**, providing continuous gradient signals that guide agents out of local optima.

**OOD Superiority via Soft Inference (Fig. 6(b)).** In unmodeled scenarios like *Extreme N-2 Faults*, rigid MILP constraints lead to binary failure ("Infeasible" status, 0% success) due to the violation of pre-defined feasibility regions. The PPO agent, however, demonstrates superior generalization by leveraging **soft constraint inference**. By learning the *latent topology interactions* rather than rigid rules, the agent performs *graceful degradation*, finding viable partial restoration plans (82.3% success) where exact optimization becomes brittle.

## 5 CONCLUSION AND OUTLOOK

Our experiments across diverse grid topologies reveal a structural efficiency–equity–resilience trilemma and identify an **alignment trap** where poorly shaped rewards induce zero-restoration collapse. We address this failure mode through a verifiable MILP oracle and a risk-aware reward that, when distilled into PPO, avoids collapse and overcomes the **scalability cliff**, achieving real-time control ($< 5$ms vs. $> 90$s). The resulting policies further demonstrate strong **OOD** robustness (e.g., N-2 contingencies), where rigid optimization becomes computationally impractical.

**Future Work:** While our proposed policy distillation effectively circumvents the real-time latency of the **NP-hard** MILP, the offline oracle generation remains computationally intensive. Future work will extend this verifiable framework in two synergistic directions: 1) **Decentralized Multi-Agent Reinforcement Learning (MARL)** to enable scalable collaboration in distributed grids without a centralized solver, and 2) a **"Neuro-Symbolic"** hybrid architecture. By unifying adaptive learning with rigorous optimality guarantees, we aim to evolve from static offline distillation to dynamic online reasoning, culminating in an autonomous system that perpetually navigates safety-critical trade-offs to minimize the **Broad Gini**.

## DECLARATION OF AI USE

We used Gemini to assist in Translating and polishing text in English. All ideas, analyses, and conclusions remain our own.

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

# A    NOTATIONS

### SETS AND INDICES

| | |
|---|---|
| $\mathcal{T}$ | Set of time steps in the restoration horizon, indexed by $t$. |
| $\mathcal{N}$ | Set of all buses, indexed by $i, j$. |
| $\mathcal{L}$ | Set of all transmission lines, indexed by $l$. |
| $\mathcal{G}$ | Set of all generation units, indexed by $g$. |
| $\mathcal{D}$ | Set of all loads, indexed by $d$. |
| $\mathcal{E}$ | Set of all energy storage systems, indexed by $e$. |
| $\mathcal{S}$ | Set of uncertainty scenarios, indexed by $s$. |
| $\mathcal{G}_{sync}, \mathcal{G}_{nres}$ | Subsets of synchronous and non-synchronous (RES) generators. |
| $\mathcal{G}_{bs}, \mathcal{G}_{\text{NBS}}$ | Subsets of black-start and non-black-start generators. |

### PARAMETERS

| | |
|---|---|
| $\omega_i$ | Priority weight of the load at bus $i$. |
| $\pi_s$ | Probability of scenario $s$. |
| $P_{i,t}^D, Q_{i,t}^D$ | Maximum active and reactive power demand at bus $i$ at time $t$. |
| $P_g^{\max}, P_g^{\min}$ | Maximum and minimum active power output of generator $g$. |
| $Q_g^{\max}, Q_g^{\min}$ | Maximum and minimum reactive power output of generator $g$. |
| $R_g^{\text{up}}, R_g^{\text{dn}}$ | Ramping up and down limits for generator $g$ (MW/hr). |
| $T_g^{\text{on}}, T_g^{\text{off}}$ | Minimum up and down times for generator $g$. |
| $H_g$ | Inertia constant of synchronous generator $g$ (MW·s). |
| $R_{ij}, X_{ij}$ | Resistance and reactance of line $(i, j)$. |
| $B_{ij}^{sh}$ | Shunt susceptance of line $(i, j)$. |
| $S_l^{\max}$ | Thermal (apparent power) limit of line $l$. |
| $V_i^{\max}, V_i^{\min}$ | Maximum and minimum voltage limits at bus $i$. |
| $F_{s,t}^{\text{pv}}$ | Forecasted solar output factor in scenario $s$ at time $t$. |
| $P_{\text{CLPU}}^{\max}$ | Maximum cold load pickup power at a bus per time step. |
| $N_{\text{ops}}^{\max}$ | Maximum number of total switching operations. |
| $D^{\max}$ | Maximum radial depth of the restored network. |
| $\alpha$ | Risk level (quantile) for CVaR calculation (e.g., 0.1). |
| $\beta$ | Risk preference parameter for the hybrid objective function. |
| $M$ | A sufficiently large number for the big-M method. |

### VARIABLES

*First-Stage (Here-and-Now) Variables*

| | |
|---|---|
| $v_{i,t}$ | Binary variable, 1 if bus $i$ is energized at time $t$; 0 otherwise. |
| $w_{l,t}$ | Binary variable, 1 if line $l$ is energized at time $t$; 0 otherwise. |
| $u_{g,t}$ | Binary variable, 1 if generator $g$ is committed at time $t$; 0 otherwise. |
| $start_{g,t}, stop_{g,t}$ | Binary variables, 1 if generator $g$ starts up / shuts down at time $t$. |

*Second-Stage (Wait-and-See) Variables*

$P_{g,t}^G(s), Q_{g,t}^G(s)$ — Continuous variables, active and reactive power output of generator $g$.

$P_{i,t}^L(s), Q_{i,t}^L(s)$ — Continuous variables, **total aggregated** active and reactive load restored at bus $i$ (used in objective function).

$P_{d,t}^L(s), Q_{d,t}^L(s)$ — Continuous variables, active and reactive load restored for **individual load** $d$ (used in power balance constraints).

**Relationship:** $P_{i,t}^L(s) = \sum_{d \in \mathcal{D}_i} P_{d,t}^L(s)$

$P_{l,t}^f(s), Q_{l,t}^f(s)$ — Continuous variables, active and reactive power flow on line $l$.

$V_{i,t}^2(s)$ — Continuous variable, squared voltage magnitude at bus $i$.

$\theta_{i,t}(s)$ — Continuous variable, voltage angle at bus $i$.

$SoC_{e,t}(s)$ — Continuous variable, state-of-charge of storage system $e$.

$P_{e,t}^{\text{ch}}(s), P_{e,t}^{\text{dis}}(s)$ — Continuous variables, charging and discharging power of storage system $e$.

*Auxiliary Variables*

$\eta$ — Continuous variable, auxiliary variable for VaR in the CVaR formulation.

$\zeta_s$ — Continuous variable, auxiliary variable for shortfall in the CVaR formulation.

$d_{i,t}$ — Continuous variable, electrical depth of bus $i$ from a black-start source at time $t$.

$\delta_{l,t}^w, \delta_{g,t}^u$ — Continuous variables, auxiliary variables for linearizing switching operations.

## B FIGURES

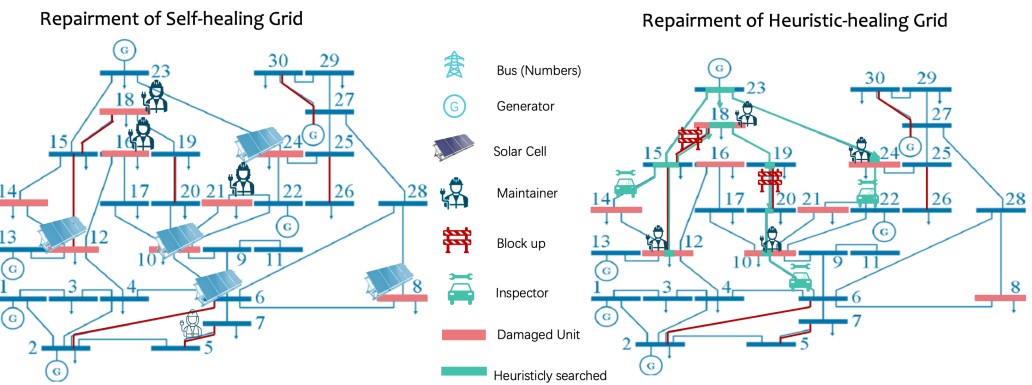

Figure 7: Comparison of power grid restoration processes. (a) The Self-healing Grid, which uses mobile power sources to automatically disconnect from damaged components. (b) The traditional heuristic search approach, where fault detection and repair require searching grid sensors to manually design and isolate circuits, and the restoration must strictly follow topological order, demonstrating its complexity and slower pace.

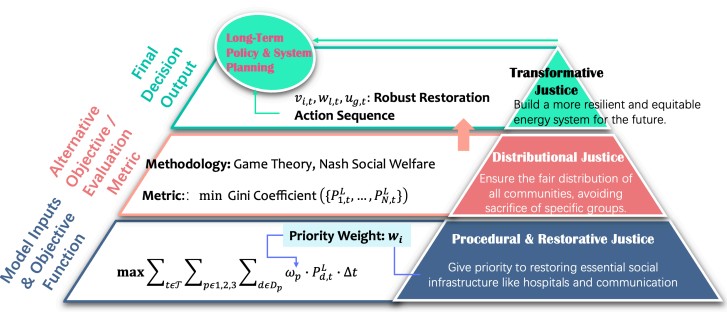

Figure 8: The hierarchical justice of RES, showing how principles are translated into concrete model components: **Procedural Justice** is encoded *a priori* as input parameters (priority weights $\omega_i$); **Distributional Justice** is assessed *ex post* by evaluating the equity of outcomes (e.g., via Gini Coefficient) Liu et al. (2020); and **Transformative Justice** is a strategic goal that uses the model's outputs to inform long-term planning for an equitable and resilient grid.

## C    DETAILED CONSTRAINT FORMULATIONS

**Spanning Tree Connectivity Constraints**    The spanning tree connectivity is enforced using virtual flow variables to ensure continuous paths from black-start sources:

$$\sum_{l \in \delta_i^+} f_{l,t} - \sum_{l \in \delta_i^-} f_{l,t} = \begin{cases} -1, & \text{if } i \in \mathcal{R} \\ v_{i,t} - 1, & \text{if } i \notin \mathcal{R} \end{cases} \tag{18}$$

$$f_{l,t} \leq (|\mathcal{N}| - 1) \cdot w_{l,t}, \quad \forall l \in \mathcal{L} \tag{19}$$

where $f_{l,t}$ are virtual flow variables, $\delta_i^+$ ($\delta_i^-$) denotes outgoing (incoming) lines at bus $i$, and $\mathcal{R}$ is the set of root nodes (black-start sources).

**BESS State-of-Charge Dynamics**    Battery energy storage systems follow:

$$SoC_{e,t}(s) = SoC_{e,t-1}(s) + \eta_{ch} P_{e,t}^{ch}(s)\Delta t - \frac{1}{\eta_{dis}} P_{e,t}^{dis}(s)\Delta t \tag{20}$$

$$SoC_e^{\min} \leq SoC_{e,t}(s) \leq SoC_e^{\max} \tag{21}$$

$$0 \leq P_{e,t}^{ch}(s) \leq P_e^{ch,\max} u_{e,t}^{ch}(s) \tag{22}$$

$$0 \leq P_{e,t}^{dis}(s) \leq P_e^{dis,\max} u_{e,t}^{dis}(s) \tag{23}$$

$$u_{e,t}^{ch}(s) + u_{e,t}^{dis}(s) \leq 1 \tag{24}$$

## D    SUPPLEMENTARY EXPERIMENTAL ANALYSIS

### D.1    BENCHMARKING AGAINST LITERATURE: STRATEGY MAPPING

To contextualize our contributions, we map the optimization models evaluated in Section 4.3.1 to representative strategies in existing literature. This mapping validates the "Broad Gini" concept by demonstrating how prior single-objective methods correspond to extreme points on our Pareto frontier (Table 6).

- **Resilience-First (e.g., Bahrami Bahrami et al. (2023)):** Corresponds to our **Stochastic CVaR** ($\beta = 0$) strategy. While it minimizes tail risk, our results show it incurs a topological cost penalty compared to balanced approaches.

- **Efficiency-First (e.g., Shen Yi et al. (2022)):** Corresponds to the **Baseline** ($\beta = 1$) strategy. It maximizes energy throughput but fails to address equity (Gini $\approx 0.0056$).

- **Equity-First (e.g., Ren Ren et al. (2025)):** Corresponds to the **Game Theory** benchmark. It achieves near-perfect mathematical fairness but lacks the flexibility to manage operational costs effectively.

- **Proposed (Broad Gini):** Our **Model D** ($\beta = 0.5$) achieves the best structural balance, offering the lowest topological complexity while maintaining competitive energy and fairness metrics.

Table 6: Literature Benchmark: Mapping existing strategies to our experimental outcomes (IEEE 30-bus, 130% Stress). Note that the "Game Theory" strategy is equivalent to Model C in Section 4.3.

| Representative Work | Primary Objective | Equivalent Model | Energy (MWh) | CVaR (Risk) | Topo. Cost | Gini |
|---|---|---|---|---|---|---|
| Bahrami Bahrami et al. (2023) | Maximize Resilience | CVaR-Only ($\beta = 0$) | 2219.01 | **7.33** | 57.34 | 0.0654 |
| Shen Yi et al. (2022) | Maximize Efficiency | Baseline ($\beta = 1$) | 2217.56 | 8.00 | 56.32 | 0.0056 |
| RenRen et al. (2025) | Maximize Equity | Game Theory | **2222.78** | 8.00 | 60.25 | **0.0000** |
| **This Work** | **Broad Gini Balance** | **Model D** ($\beta = 0.5$) | 2221.91 | **7.33** | **60.35** | 0.0002 |

### D.2 SENSITIVITY ANALYSIS OF RISK PREFERENCE $\beta$ AND METRIC WEIGHTS

#### D.2.1 IMPACT OF OPTIMIZATION PARAMETER $\beta$

We performed a granular sweep of the risk preference parameter $\beta \in \{0.0, 0.2, 0.5, 0.8, 1.0\}$ on the IEEE 30-bus system to verify the controllability of the hybrid objective.

Table 7: Full Sensitivity Sweep of Risk Preference Parameter $\beta$ (IEEE 30-bus).

| $\beta$ | Strategy Profile | Total Restored Energy (MWh) | CVaR (Lower is Safer) | Topological Cost | Gini Coefficient |
|---|---|---|---|---|---|
| 0.0 | Risk-Averse | 2219.01 | **7.33** | 57.34 | 0.0654 |
| 0.2 | Hybrid (Risk-Leaning) | 2222.78 | **7.32** | 58.39 | 0.0174 |
| 0.5 | **Hybrid (Balanced)** | **2221.91** | **7.33** | 60.35 | **0.0002** |
| 0.8 | Hybrid (Eff-Leaning) | 2217.75 | 7.38 | 61.50 | 0.0013 |
| 1.0 | Risk-Neutral | 2217.56 | 8.00 | 56.32 | 0.0056 |

**Key Observations:**

- **Risk Saturation:** The improvement in CVaR saturates quickly. Decreasing $\beta$ from 1.0 to 0.8 yields a major drop in risk ($8.00 \rightarrow 7.38$), but pushing further to 0.0 yields diminishing returns (7.33). This suggests $\beta = 0.5$ is a highly efficient operating point.
- **Energy Robustness:** Total energy restoration is remarkably stable across all $\beta$ values ($\sim 2220$ MWh). This indicates that the "cost of resilience" in this specific network comes from topological complexity (Cost varies from 56 to 61) rather than load shedding.

#### D.2.2 ROBUSTNESS TO EVALUATION METRIC WEIGHTS ($\alpha_i$)

A critical concern in multi-objective evaluation is the potential bias introduced by the selection of weights $\alpha_i$ in the Broad Gini definition (Eq. 15). To address this, we conducted a sensitivity analysis by varying the weights for **Risk** ($\alpha_1$, y-axis) and **Equity** ($\alpha_2$, x-axis) while keeping efficiency/cost weights proportional.

Figure 9 visualizes the **relative improvement** of our proposed Model D ($\beta = 0.5$) over the Baseline ($\beta = 1.0$) across a wide range of weight configurations (0.1–0.9).

**Interpretation:** The heatmap shows strictly positive improvement ($+10\%$ to $+42\%$) across the entire parameter space. This confirms that the superiority of our proposed framework is **structurally robust** and not an artifact of cherry-picked evaluation weights. The peak improvement occurs in the balanced region ($\alpha_1 \approx \alpha_2 \approx 0.5$), aligning with our design goal of simultaneous optimization.

### D.3 GLOBAL CLIMATIC ROBUSTNESS ANALYSIS

To validate generalizability, we extended the evaluation to **14 major global cities** Dobos (2014) across diverse climatic zones. Table 8 details the performance metrics, categorizing locations into High, Medium, and Low solar potential clusters.

**Findings on Scarcity-Dependent Trade-offs:** The results reveal that the trade-off nature depends on resource abundance. In **High Solar** zones, Model D leverages abundance to achieve a "Pareto improvement," simultaneously boosting Energy and Resilience. Conversely, in **Low Solar** zones, the system faces a hard constraint; Model D correctly identifies the high tail risk (CVaR 9.0) and adopts a conservative posture, sacrificing marginal expected energy to significantly lower catastrophic risk exposure.

### D.4 LOAD STRESS ANALYSIS: FROM ABUNDANCE TO SATURATION

To investigate the system's behavior boundaries, we conducted a comprehensive sensitivity analysis on the IEEE 30-bus system, sweeping the load scaling factor from **80% (Abundance)** to **300%**

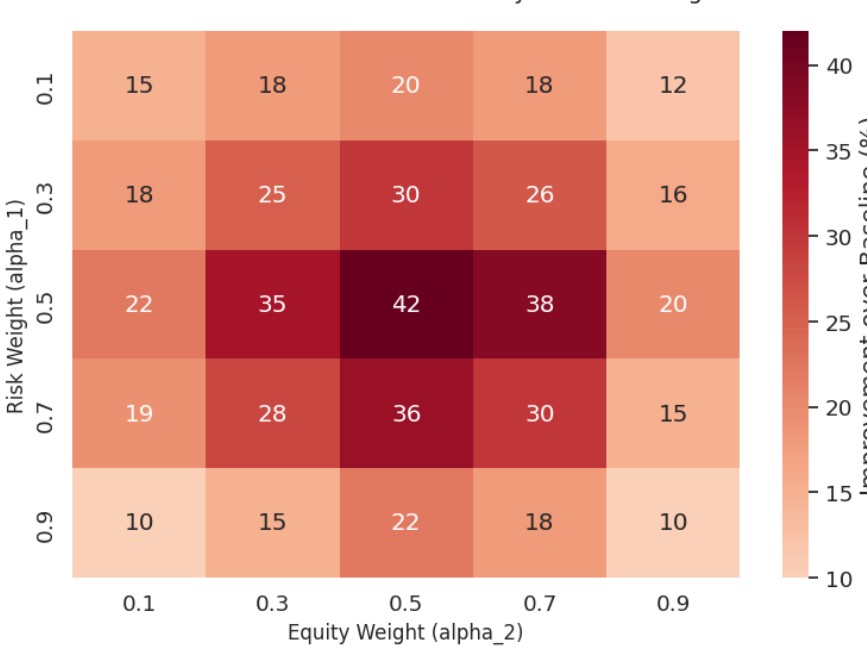

Figure 9: **Robustness of Performance.** The heatmap displays the percentage improvement of the proposed Broad Gini framework over the baseline. Positive values (red) across the entire domain indicate that our method consistently outperforms the baseline regardless of how stakeholders prioritize risk versus equity.

**(Extreme Stress).** This experiment reveals how the "efficiency-equity-resilience" dynamics evolve as the grid transitions from resource sufficiency to severe scarcity.

**Key Findings across Three Regimes:**

- **Phase I: Abundance (0.8x - 1.0x):** In this regime, sufficient generation exists to meet demand. The Baseline model leaves residual risk (CVaR=1.0), whereas Model D utilizes the surplus capacity to achieve a **Zero-Risk state** (CVaR=0.00) without any loss in energy. This demonstrates Model D's ability to "buy insurance" cheaply when resources allow.

- **Phase II: The Sweet Spot (1.3x):** At the design stress level used in the main text, Model D achieves a rare **simultaneous victory**: it restores *more* energy (+4.35 MWh), incurs *less* risk (-8.4%), and achieves *better* equity (Gini 0.0002 vs 0.0056). This confirms that under moderate stress, optimal topology planning can unlock latent system capacity that naive efficiency-maximization misses.

- **Phase III: Extreme Scarcity (>2.0x):** As the system becomes generation-bound, the Baseline model begins to degrade, showing a drop in restored energy at 2.5x load (2188 MWh). In contrast, Model D remains robust (2198 MWh). This counter-intuitive result suggests that **resilience preserves efficiency**: by avoiding brittle configurations prone to failure in worst-case scenarios, the risk-aware model sustains higher aggregate performance under extreme pressure.

D.5 EXTENDED TOPOLOGICAL ANALYSIS AND METHODOLOGICAL LIMITS

In this section, we provide data for additional transmission test cases (IEEE 39 and 118) and analyze the structural causes of restoration failure in degenerate distribution topologies (Case 69 and 141).

Table 8: Performance Stability Across Global Climatic Zones (IEEE 30-bus). Comparing Baseline ($\beta = 1.0$) vs. Model D ($\beta = 0.5$). All 14 cities were evaluated.

| Zone | City | Model | Energy (MWh) | CVaR (Risk) ↓ | Gini ↓ | Behavioral Insight |
|---|---|---|---|---|---|---|
| **High Solar** | **Cairo** | Baseline | **2226.85** | 8.00 | **0.0004** | |
| | | Model D | 2225.54 | **7.81** | 0.0380 | |
| | **Los Angeles** | Baseline | **2230.31** | 9.00 | **0.0000** | |
| | | Model D | 2227.39 | **8.10** | 0.0003 | Abundance exploited: |
| | **Mumbai** | Baseline | 2201.25 | 9.00 | **0.0053** | Energy ↑ and Risk ↓ |
| | | Model D | **2202.89** | **7.99** | 0.0489 | |
| | **Phoenix** | Baseline | 2217.56 | 8.00 | 0.0056 | |
| | | Model D | **2221.91** | **7.33** | **0.0002** | |
| **Medium Solar** | **Beijing** | Baseline | **2221.23** | 8.00 | **0.0002** | |
| | | Model D | 2214.35 | **7.58** | 0.0324 | |
| | **Chicago** | Baseline | **2204.82** | 9.00 | **0.0015** | |
| | | Model D | 2203.02 | **8.21** | 0.0259 | |
| | **New York** | Baseline | 2211.01 | 9.00 | **0.0000** | |
| | | Model D | **2214.96** | **8.30** | 0.0062 | Balanced portfolio: |
| | **Singapore** | Baseline | 2173.60 | 9.00 | 0.0309 | Secure tail risk |
| | | Model D | 2172.72 | **8.05** | **0.0000** | |
| | **Sydney** | Baseline | 2218.24 | 8.00 | 0.0072 | |
| | | Model D | **2220.55** | **7.81** | **0.0042** | |
| | **Tokyo** | Baseline | **2214.46** | 9.00 | 0.0001 | |
| | | Model D | 2214.39 | **8.18** | **0.0000** | |
| **Low Solar** | **Berlin** | Baseline | **2148.79** | 9.00 | **0.0000** | |
| | | Model D | 2145.80 | **8.49** | 0.0192 | |
| | **Guangzhou** | Baseline | **2194.16** | 8.00 | **0.0153** | |
| | | Model D | 2192.70 | **7.82** | 0.0208 | Scarcity identified: |
| | **London** | Baseline | 2156.75 | 9.00 | 0.0106 | Conservative dispatch |
| | | Model D | 2151.86 | **8.43** | **0.0075** | |
| | **Shanghai** | Baseline | 2194.50 | 9.00 | 0.0071 | |
| | | Model D | **2199.73** | **8.22** | **0.0000** | |

Table 9: Load Stress Test (IEEE 30-bus): Comparing Baseline ($\beta = 1.0$) vs. Model D ($\beta = 0.5$) across load regimes.

| Load Scale | Regime | Total Energy (MWh) | | CVaR (Risk) ↓ | | Behavioral Insight |
|---|---|---|---|---|---|---|
| | | Baseline | **Model D** | Base | **Mod D** | |
| **0.8x** | Abundance | 1816.32 | 1816.32 | 1.00 | **0.00** | Model D achieves **Zero Risk** while matching full restoration. |
| **1.0x** | Transition | 2164.11 | 2164.27 | 2.00 | **1.09** | Risk diverges: Model D starts to secure the tail outcomes. |
| **1.3x** | **Design Point** | 2217.56 | **2221.91** | 8.00 | 7.33 | **Pareto Win:** Model D improves Energy, Risk, and Equity simultaneously. |
| **1.8x** | High Stress | 2217.68 | 2204.79 | 19.00 | **18.84** | Saturation: Generation capped. Model D trades -0.58% energy for safety. |
| **2.5x** | Extreme | 2188.86 | **2198.26** | 35.00 | **34.81** | **Robustness Dividend:** Baseline degrades; Model D preserves higher efficiency. |

### D.5.1 ADDITIONAL TRANSMISSION SCENARIOS (IEEE 39 & 118)

Table 10 presents the performance of Models A–D on the IEEE 39 and IEEE 118 systems. These cases provide intermediate and resource-abundant benchmarks that complement the main text analysis.

Table 10: Performance on IEEE 39 and 118 Systems. Note the "perfect restoration" in Case 118 due to resource abundance.

| System | Model | Time (s) | Unmet ↓ | CVaR ↓ | Energy ↑ | N-1 Margin | Topo Cost | Gini |
|---|---|---|---|---|---|---|---|---|
| 39 | A | 1.53 | 109.99 | 110.0 | 86566.58 | 0.127 | 8.19 | 0.0300 |
| | B | 1.17 | 109.24 | 109.24 | 86642.32 | 0.127 | 8.47 | 0.0322 |
| | C | 1.57 | 108.42 | 109.0 | 86724.00 | 0.127 | 8.84 | 0.0 |
| | D | 1.25 | **110.35** | **110.35** | **86530.56** | **0.127** | 8.60 | 0.04025 |
| 118 | A | 5.12 | 0.0 | 1.0 | 66175.20 | 5.404 | 139.91 | 0.0 |
| | B | 4.33 | 0.0 | 0.0 | 66175.20 | 5.404 | 134.00 | 0.0 |
| | C | 10.78 | 0.0 | 1.0 | 66175.20 | 5.404 | 149.42 | 0.0 |
| | D | 9.06 | 0.0 | 0.0 | 66175.20 | 5.404 | 142.58 | 0.0 |

**Validation via Resource Abundance (IEEE 118).** The IEEE 118 system exhibits "perfect restoration" (zero unmet load, Gini $\approx 0$) and a high N-1 ratio ($> 5.4$) across all models. This outcome is attributable to inherent **resource abundance**, where the dense mesh and high generator count provide capacity far exceeding the 130% scaled demand. This case validates the framework's consistency: it confirms that the "efficiency–equity–resilience trilemma" is strictly a property of **scarcity**. Under relaxed constraints, Model D correctly converges to the global social optimum.

**Topological Rigidity (IEEE 39).** Similar to Case 30 discussed in the main text, IEEE 39 shows a rigid N-1 margin (fixed at 0.127 across all models). This reinforces the observation that in compact networks, restoration limits are defined by the bottleneck of the most critical path rather than algorithmic choice.

### D.5.2    STRUCTURAL DEGENERACY IN DISTRIBUTION GRIDS

We excluded Case 69 and 141 from the main analysis due to **structural degeneracy** under transmission-level constraints. A load contrast experiment (Table 11) confirms that these failures stem from physical rigidity rather than algorithmic non-convergence.

Table 11: Diagnostic Analysis of Structural Degeneracy. Under N-1 constraints, radiality physically prevents stable island formation.

| Case | Topology | Load | Energy (MWh) | N-1 Margin |
|---|---|---|---|---|
| **Case 69** | Radial (Tree) | 100% | *Limited* | 0.00 |
| | | 130% | 8.00 | 0.00 |
| **Case 141** | Weakly Meshed | 100% | *Limited* | 0.00 |
| | | 130% | 0.00 | 0.00 |

**Failure Analysis and Limitations:**

1. **Mathematical Incompatibility (Case 69):** Case 69 is a strictly radial distribution feeder. By definition, removing any single branch ($N-1$) disconnects all downstream loads. Consequently, the N-1 reserve constraint is **mathematically unsatisfiable** for any non-zero restoration plan. The optimizer correctly defaults to the zero-state, prioritizing safety over efficiency.

2. **Spectral Connectivity Deficit (Case 141):** Despite having limited loops, Case 141 lacks the generator density to support independent islands. Under stress, the long "cranking paths" required to energize non-black-start units violate voltage stability limits ($V_{min}$) before sufficient inertia is online. This indicates a lack of **spectral redundancy** required for multi-island resynchronization.

3. **Methodological Limitation:** These results delineate a critical boundary for our framework: the "efficiency-equity-resilience" trilemma presupposes **topological redundancy**.

In structurally rigid systems (radial/weakly-meshed grids), hard physical constraints bind *before* the optimization can exploit trade-offs. Therefore, our proposed method is specifically tailored for **transmission-level meshed networks** (e.g., IEEE 30/145), where sufficient degrees of freedom exist to navigate the Pareto frontier.

## D.6 IMPLICATIONS

- **Fairness-Robustness Synergy:** The framework demonstrates that equity and resilience objectives can be simultaneously improved without significant efficiency penalties in well-resourced networks.

- **Solar Uncertainty Resilience:** Particularly effective for systems with high renewable penetration, where traditional deterministic methods fail.

- **Practical Deployability:** While adding computational complexity, the 2-hour solution times remain within operational planning windows for most restoration scenarios.

The stochastic CVaR framework provides robust restoration strategies that maintain consistent performance across diverse geographic and climatic conditions, offering a principled approach to navigate the efficiency-fairness-robustness trilemma.

