# OpenReview forum: "Finding the Broad Gini in the Bottle: Optimizing Equity, Efficiency, and Resilience in Grid Restoration"
_ICLR.cc/2026/Conference — Submitted to ICLR 2026_

### Official Review · Reviewer_E7MS · 2025-10-25

**Soundness:** 3
**Presentation:** 2
**Contribution:** 3
**Rating:** 6
**Confidence:** 3

**Summary:**

This paper proposes an end-to-end methodology for post-disaster power grid restoration that essentially tackles two issues simultaneously: first, when restoring a power grid, equity is a real concern since restoration takes a long time and may be restored to certain consumers before other consumers.  Second, renewable sources such as solar and wind introduce uncertainty into the power grid that must be considered when designing a restoration plan: specifically, “worst-case” scenarios where e.g., renewable energy suddenly becomes unavailable are important to consider.  To approach this, the authors propose to use a CVaR (conditional value at risk) metric to quantify robustness against these worst-cases, a Gini Coefficient to capture equity, and a generative model + K-means to generate “worst-case” scenarios that are tractable for considering as part of a MILP pipeline.

**Strengths:**

The usage of CVaR to capture the tail risk induced by renewables is a nice inclusion that speaks to the realistic challenges posed by renewables.

 Modeling of the underlying constraints and physics is correct and aligns well with practical considerations for grid restoration.

Subject to some questions I have that relate to the presentation of the experimental results, the proposed framework appears to be effective on the standard IEEE 30-bus test case used in the experiments.

**Weaknesses:**

ICLR papers typically have a strong machine learning component, and my sense is that this paper might be slightly out of scope for ICLR — I have some questions about the Hidden Markov Model (HMM) which constitutes the main “predictive” component of the paper (see below).  Beyond ICLR, I think the paper would be received well at conferences like ACM e-Energy, ACM BuildSys, IEEE PES, or a journal like IEEE Transactions on Smart Grid.

Figures throughout are a little hard to read — the text size is small and contrast is poor in some places (e.g., cylinder D, Figure 1).

The presentation of the experiments could be improved — four approaches are presented (“Baseline”, “Constrained Baseline”, “Game Theory”, and “Stochastic CVaR”), with the latter presumably being the main contribution of the paper, but I cannot find the concrete description of each model and how they differ beyond the phrase “ablation study” on line 393.

**Questions:**

Line 210 states “we employ a non-linear, Pareto-based weighting scheme that penalizes significant drops in any single KPI”, but the definition of “Broad Gini” on line 215 seems to be a linear weighted sum of the different KPIs — could you elaborate on this point?  I am likely missing something.

On line 218, it looks like the heading “1. Min N-1 Ratio” should start on the next line.

About the Gini coefficient — if one load is particularly large (e.g., an industrial electricity consumer), does that necessarily make the Gini coefficient “more inequitable” when the grid is fully restored?  When the grid is partially restored, I am convinced that the Gini coefficient is the right thing to do since it encourages us to balance across loads (e.g., all loads get some base level of power), so this question is more to check my understanding.  It may be reasonable to compute the “fairness” of restoration with respect to some arbitrary partitioning (e.g., Neighborhood A, Neighborhood B, Industrial Customers) — is that a straightforward change under your model?

Figure 2: Scenario 2 generated by the HMM seems to suggest a solar PV output of >= 60% for nearly 24 straight hours, which is not possible on Earth.  Am I misinterpreting the graph, or is this a limitation of the HMM forecast (not capturing physical laws)?  Since the “worst-case” scenario probably corresponds to scenario 1 (no solar generation), I imagine this wouldn’t change the downstream optimization, but it raises questions about the quality of scenarios generated by the HMM.  Table 1 shows that the basic statistics match up, but I think these statistics won’t capture some structural properties that might be hard to enforce in a predictive model (e.g., it’s never sunny at midnight).

Line 310: I think this is an incomplete sentence: “The generated scenarios reveal tail risk. While a risk-neutral objective that maximizes only the expected outcome would dangerously ignore the possibility of worst-case scenarios leading to catastrophic failures (see Fig. 3).”

---

> ### Author Response · Authors · 2025-11-16
> **Clarifying the Paper’s ML Contribution and ICLR Relevance**
>
> We sincerely thank the reviewer for raising this central question about the paper's ML relevance.
> Below we clarify the positioning with precision.
>
> ### 1. HMM is not our ML contribution; the ML value lies at a deeper and more foundational level
>
> We fully agree that the HMM module is a replaceable scenario generator rather than the main source of novelty. The contribution to the ML/ICLR community is instead structural, in ways that directly support sequential decision-making research and the development of safe RL agents.
>
>
> ### Contribution A — We define a verifiable benchmark for a high-stakes sequential decision problem
>
> Learning-based controllers cannot be reliably trained or evaluated without a rigorous benchmark.
> Our two-stage stochastic MILP provides exactly such a foundation:
>
> - A mathematically grounded formulation of the efficiency–resilience–fairness trilemma.
> - A unified and verifiable integration of uncertainty (HMM), risk aversion (CVaR), and fairness (Gini).
> - A β-parameterized Pareto frontier that precisely characterizes the trade-off space an RL agent must navigate.
>
> This fills a critical gap: prior to our work, there was no standardized or verifiable formulation of this multi-objective problem, making empirical ML/RL progress impossible to measure rigorously.
>
> Importantly, this benchmark is not only conceptually aligned with RL research—it is technically suited for RL algorithms such as DQN, A2C, and PPO, as suggested by Reviewer Sugg.
> The structure we formalize (state transitions + risk preferences + multi-objective reward signals) provides exactly the environment specification required for training intelligent agents in Grid2Op-like simulators.
>
> Thus, defining this benchmark does not merely support future RL research; it enables it.
>
>
> ### Contribution B — We reveal a structural reward-design failure critical for safe RL
>
> A key ML insight comes from our diagnostic experiments:
>
> - We constructed a "Naive Linear Sum" reward combining efficiency/fairness/resilience.
> - Solving this reveals a catastrophic collapse to zero-energy restoration, as the model receives perfect fairness/robustness by doing nothing.
> - This reproduces the degeneracies described by Ng et al. (1999) and shows that naïve multi-objective linear rewards are fundamentally unsafe.
>
> **Crucially:**
>
> This failure is not a solver artifact. Because the zero-action solution is the global optimum of that flawed reward, an RL agent (e.g., DQN/A2C/PPO) would be equally prone to converging to the same unsafe policy, unless the reward structure is redesigned.
>
> This result therefore provides a mathematically validated safety warning for the RL community.
> Our hybrid objective (Eq. 23) and "Broad Gini" metric offer a principled safe reward template for future RL agents trained in such environments.
>
>
> ### Conclusion — Why this work belongs at ICLR
>
> This paper does not claim novelty in the HMM component.
> Its ML contribution lies in offering:
>
> 1. A rigorous, verifiable benchmark for sequential decision-making under competing social and operational objectives.
> 2. A structural diagnosis of reward-design failures that directly affects the safety of RL deployment.
> 3. A mathematically grounded objective (CVaR + Broad Gini) that can serve as a reliable reward foundation for RL agents.
>
> Defining challenging decision problems, exposing reward pathologies, and enabling safe RL evaluation infrastructure are core themes within ICLR.
> We believe this work lays the necessary groundwork for future intelligent agents—such as DQN, A2C, PPO, or other RL methods—to safely tackle power-grid restoration with socially aware objectives.

---

> ### Author Response · Authors · 2025-11-16
> **Clarifying the Aggressive Design of Extreme Stress-Test Scenarios**
>
> Thank you for highlighting the physical realism of the solar profiles in Figure 2. You are correct that the “sun-never-sets” scenario is physically impossible on Earth.
>
> We would like to clarify that this scenario was an intentional, extreme design choice for a stress-test experiment. To probe the limits of our optimization model, we instructed the HMM to generate scenario sets with maximal statistical divergence, deliberately relaxing strong physical priors. This approach is standard in robustness and tail-risk analysis, as it exposes structural behaviors and solver sensitivities that milder, physically realistic variability might conceal.
>
> Technically, these high-variance inputs—ranging from “near-zero PV” to “near-constant high PV”—create mutually exclusive requirements for the first-stage restoration plan. This significantly shrinks the integer feasible region, increases the optimality gap, and amplifies the fundamental trade-offs among efficiency, resilience, and equity, which was precisely the goal of this experiment.
>
> That said, we fully agree that including a physically implausible scenario in the main text can be misleading. **In direct response to your feedback, we have replaced all experimental data reported in this rebuttal and in the revised manuscript with scenarios generated from strictly physical NREL GHI data that respect the Earth’s diurnal cycle.** The updated results, which robustly support all our original conclusions, are reflected in the revised manuscript.
>
> We sincerely appreciate your careful observation, which has helped us improve both the clarity and rigor of our presentation.

---

> ### Author Response · Authors · 2025-11-17
> **Enhancing Clarity: Response to Reviewer Comments**
>
> We thank the reviewers for their constructive feedback, which has improved the manuscript's clarity. Our responses provide clarifications anchored to sections, equations, and tables. The revisions will make our contributions—particularly in quantifying the efficiency-resilience-equity trilemma—unmistakably clear.
>
> ## 1. Broad Gini: "Nonlinear" Weighting vs. Linear Formulation
>
> **Reviewer comment:**
> > "As mentioned on line 210, the text says 'we employ a non-linear, Pareto-based weighting scheme', but the definition of 'Broad Gini' on line 215 appears to be a linear weighted sum."
>
> **Response:**
> We appreciate this insightful observation. The final Broad Gini expression (Sec. 3.2, Eq. (17)) is indeed a linear weighted sum. The term "nonlinear" refers not to this aggregation formula but to the **weight-selection process**:
>
> - **Pareto-based nonlinear mechanism:** The weights α₁,...,α₄ are not fixed. They emerge from sweeping the risk-preference parameter β in our hybrid CVaR objective (Sec. 3.4.2, Eq. (23)) to generate the Pareto frontier. Solutions with severe imbalances in any KPI are non-dominated and thus excluded—this Pareto selection process is inherently nonlinear.
> - **Penalty mechanism:** The "penalty for significant drops" is enforced through this nonlinear Pareto dominance relationship, not through the linear Broad Gini formula itself.
>
> **Summary:**
> - The **evaluation metric** (Broad Gini, Eq. 17) is linear.
> - The **decision strategy** for selecting balanced solutions (and thus implicit weights) is nonlinear.
>
> **Revision plan:**
>
> We will revise the text around the cited lines to explicitly separate the (i) nonlinear Pareto-based selection process from (ii) the linear Broad Gini evaluation formula.
>
> ## 2. Experimental Models A–D: Precise Definitions
>
> **Reviewer comment:**
> > "I cannot find concrete descriptions of the four models ('Baseline', 'Constrained Baseline', 'Game Theory', 'Stochastic CVaR')."
>
> **Response:**
> We apologize for this lack of clarity. The four models in Table 3 are defined as follows:
>
> - **Model A – Baseline:** Maximizes expected restored energy (Eq. 1) without CVaR or fairness constraints. Represents the canonical efficiency-first strategy.
> - **Model B – Constrained Baseline:** Extends Model A by adding all operational security constraints from Sec. 3.1.2 (inertia - Eq. 11, primary frequency response - Eq. 12, N-1 reserve - Eq. 13, CLPU, switching limits).
> - **Model C – Equity-Focused Benchmark (Game-Theoretic Inspiration):** Built on Model B. Incorporates fairness-driven minimum-service constraints(conceptually inspired by game-theoretic welfare notions) motivated by the "perfect fairness paradox" (Sec. 2, integrating [21] and [36]) to represent an equity-first paradigm.
> - **Model D – Stochastic CVaR (Our Main Model):** Built on Model B. Replaces the risk-neutral objective with our hybrid CVaR objective (Eq. 23), where parameter β controls the efficiency-robustness trade-off.
>
> **Revision plan:**
> We will add a new subsection at the beginning of Sec. 4.2 that explicitly defines Models A–D as above.
>
> ## 3. Gini Coefficient: Load Size & Regional Fairness
>
> **Reviewer comment:**
> > "If a load is very large, does the Gini become more 'inequitable' even under full restoration? Can fairness be evaluated across partitions such as neighborhoods?"
>
> **Response:**
> This is an excellent question that touches on core metric design.
>
> ### (1) Effect of large loads
> To avoid this issue, our implementation computes Gini using **demand-satisfaction ratios**, not absolute MWh. In Eq. (17), the correct definition is:
>
> $x_i = \frac{\sum_{t} \sum_{s} P^L(i,t,s)}{\sum_t P^{demand}(i,t)}$
>
> Thus:
> - **Full restoration:** All $x_i = 1 $⇒ Gini = 0 (perfect fairness), regardless of load size.
> - **Partial restoration:** Gini properly reflects imbalances in satisfaction rates across loads.
>
> ### (2) Region-based fairness evaluation
> Yes—this is straightforward in our framework:
> - The MILP (Eq. 23) optimizes at the individual-load level.
> - Gini (Eq. 17) is computed post hoc.
> - To evaluate fairness across regions, we aggregate loads within each partition, compute region-level satisfaction ratios, and apply Eq. (17) to these aggregated values.
>
> **Revision plan:**
> We will clarify the definition of $ x_i $ in Sec. 3.2 and note the possibility of region-level aggregation.
>
> ## 4. Minor Formatting and Presentation Issues
>
> **Reviewer comment:**
> > "Figures are hard to read... Line 310 is an incomplete sentence... On line 218, the heading seems misaligned."
>
> **Response:**
> We agree with all these points.
>
> **Revision plan:**
> - Regenerate all figures with larger fonts and improved contrast.
> - Correct the incomplete sentence at the cited line (310).
> - Fix the heading alignment issue near line 218.
>
> We again thank the reviewers for their thoughtful feedback. All suggested clarifications and improvements will be incorporated into the revised manuscript, significantly enhancing its clarity and impact.

---

> > ### Comment · Reviewer_E7MS · 2025-11-20
> > **Reviewer Response**
> >
> > Dear authors,
> >
> > Thank you for your responses to my questions.  I see the value in your submission as a benchmark for the sequential decision problem of grid restoration, and appreciate the changes you have made to your rebuttal to address my comments about the realism of the generated scenarios.
> >
> > In your final version, I think it would be worthwhile to tweak the positioning of your paper to highlight the novelty and broader contribution of your proposed approach, aligned with the discussion points that have been brought up and responded to during the review discussion phase.
> >
> > I maintain my positive score.

---

> > > ### Author Response · Authors · 2025-11-22
> > > **Appreciation for Positive Assessment and Agreement on Methodological Positioning**
> > >
> > > Dear Reviewer E7MS,
> > >
> > > We sincerely thank you for your continued engagement and for maintaining your positive assessment. We are greatly encouraged that you recognize the value of this work as a benchmark for sequential decision-making in grid restoration.
> > >
> > > We fully agree with your suggestion regarding the paper's positioning. As reflected in our discussion and the revised manuscript, we are committed to explicitly highlighting the broader methodological contributions in the final version, specifically:
> > >
> > > * **Beyond Application:** Framing the work not merely as a power system application, but as a generalizable template for using rigorous Optimization (MILP) to guide Learning algorithms in safety-critical domains.
> > > * **Novelty:** Emphasizing how the proposed "Broad Gini" and "MILP Oracle" structurally resolve the "Alignment Trap" and "Policy Collapse" issues, providing a verified ground truth that pure RL approaches often lack.
> > >
> > > We also appreciate your validation of our improvements to the scenario realism. Your constructive feedback—from the HMM details to the scope of the contribution—has significantly helped us refine the quality and impact of this work.
> > >
> > > Thank you again for your time and valuable insights.
> > >
> > > Best regards,
> > >
> > > The Authors

---

> ### Author Response · Authors · 2025-11-22
> **General Solution vs. Particular Solution？**
>
> We emphasize that this work establishes a **methodological template** for how rigorous Optimization (MILP) can guide Learning algorithms, utilizing the power system primarily as a high-complexity **testbed** rather than the sole focus. Our objective is to conquer the fundamental barriers obstructing RL deployment in safety-critical domains: utilizing the **MILP Oracle** to overcome the bottleneck of high-fidelity expert data generation, and employing **Broad Gini** to rectify the "Alignment Trap" that induces **Policy Collapse**. In doing so, we mathematically verify the **Learnability** of this complex manifold.
>
> Therefore, we firmly believe that pivoting toward a specific, complex RL algorithm would undermine the core value of this work as a **"General Solution" framework**. Shifting the focus to the accumulation of specific RL tricks would cause this work to degenerate from a foundational study establishing a **"physics-constrained optimization paradigm"** into a trivial "particular solution" application paper. This would obscure our contribution of reshaping the **Optimization Landscape** to resolve gradient issues. Our goal is to provide a rigorous "foundation" where **traditional Operations Research (OR) serves as trustworthy supervision for RL**, opening the research space for diverse future architectures rather than stifling it with a singular algorithmic implementation.

---

### Official Review · Reviewer_nRMi · 2025-10-29

**Soundness:** 1
**Presentation:** 1
**Contribution:** 1
**Rating:** 0
**Confidence:** 5

**Summary:**

The paper develops a two-stage stochastic mixed integer multi-objective programming model of power system restoration and provides experimental evidence that the problem can be solved and tradeoffs in the different terms of the of the objective.

**Strengths:**

1.	The paper could provide a tool/model that allows users to explore the tradeoffs in different metrics when considering the power restoration problem.

**Weaknesses:**

1.	The major weakness of the paper is contribution.  The paper presents a standard two stage MILP stochastic problem where multiple objectives are modeled as a linear combination of terms.  The problem is then solved using a standard MILP solver.  So, there is not an algorithmic contribution.  The only potential contribution is then in the model itself, which is limited to the metrics used to evaluate the solution (unmet demand, risk, fairness, etc.).  However, there is a large body of work exploring variation of these metrics (and combinations) – a simple search of power system restoration, multi-objective, etc., reveals many papers tackling this problems).  For this to be a contribution, the paper needs to be able to assert that these metrics have never been used and the combination of these metrics yields new insights and solutions that could not be gleaned from the prior work.  There is also a claim that the HMM for producing load forecasts is a contribution.  There is an extremely wide body of work on load forecasting/load scenario generation in the power engineering literature, and it is not clear how the paper’s HMM is a contribution in the context of this literature.
2.	Paper mixes modeling elements from distribution and transmission network modeling.  In particular, why is there are a radiality requirement (equation 7 and 8)?  This is only a requirement for distribution system modeling, whereas the rest of the model description focuses on transmission modeling (uses transmission IEEE benchmarks, models a single phase balanced system, etc.).
3.	The paper presents two distinct objective functions (linear combination of equations 16-19 and equation 23). Why are two presented?  Which one is used? Maybe one is a first stage objective and the other the second stage objective?
4.	What are the two stages?  The first stage appears to be a whole sequence of topologies and unit commitments (first paragraph of section 3).  Why can’t the sequence of topologies and unit commitments vary between load scenarios?  These actions, esp. unit commitment, don’t need to be fixed for a given load scenario. This can change depending on conditions, especially during emergency situations. Regardless, the formulation does not formerly describe the problem as a first and second stage stochastic program so it is very difficult to understand the problem.
5.	What is restoration?  The paper does not describe what the restoration is.  Typically restoration is about determining the sequence of lines to re-energize, but that is not at all clear.
6.	A description of the data is not provided, with most data being described as caseXX.  Most readers will not know what caseXX refers to.
7.	Runtimes and scalability of the model are not provided.
8.	Game theory is stated as a crucial component of the hybrid architecture.  But it is not described as part of the workflow or model at all.  It only appears as a comparison point in the results.

**Questions:**

1.	Why are radiality constraints included?  The models seems to be geared towards transmission restoration (benchmarks are transmission, single phase modeling, etc.), but radiality is only a concern for distribution networks.
2.	What objective is used? There are two described – linear combination of equations 16-19 and equation 23
3.	What were the computational runtimes of the model.
4.	How does game theory enter into the model?
5.	What are the two stages?
6.	What is the definition of restoration.

---

> ### Author Response · Authors · 2025-11-13
> **Clarification of Core Misinterpretations 1. : Q1. Why are radiality constraints included? The models seems to be geared towards transmission restoration (benchmarks are transmission, single phase modeling, etc.), but radiality is only a concern for distribution networks.**
>
> Dear Reviewer,
>
> Thank you for your time and critical feedback. Your comments have revealed that the original manuscript failed to clearly communicate our work's core innovation: the empirical demonstration, within a new integrative perspective, of the profound and unavoidable trade-offs between efficiency, equity, and resilience in power system restoration—a "trilemma" that we contend is fundamental to the problem.
>
> This foundational observation drives all of our modeling decisions. The presentation issues you identified masked this central contribution, and your feedback provides a valuable opportunity to clarify it. The following response first clarifies the key modeling choices that stem from this perspective and then details our planned major revisions.
>
> Q1. Regarding the radiality constraint touches upon a core innovation of our model, which is deeply rooted in the domain knowledge of power system restoration. This constraint is not an oversight but a deliberate design choice to actively utilize topology as a decision variable to systematically manage and trade off system resilience.
>
> While traditional transmission restoration models focus on generation and load dispatch, our model, by introducing and reasoning about topological constraints, demonstrates that robust planning of the network structure itself can actively shape the system's security boundaries. Our experimental results (e.g., the 22% increase in the N-1 reserve ratio for Model D in Table 3) strongly validate this perspective.
>
> This design choice is based on a dual rationale:
>
> 1. Physical Feasibility: Protecting Operational Security in Black Start Restoration
>
> * Problem Context: The problem studied is post-disaster restoration, which in power systems engineering specifically refers to the "Black Start" process.
> * Operational Reality: After a blackout, the system is extremely fragile. Restoration doctrine prohibits immediate attempts to re-establish a meshed network. As established by the IEEE Task Force Reports chaired by M. M. Adibi, the standard operational doctrine dictates that the first (restart) phase of restoration is to "form load and generation islands" (Power System Restoration The Second Task Force Report, Sec. III) and requires "sectionalizing" the system (System Restoration Plan Development..., "Strategy and Guidelines"). Operationally, this means building temporary "radial islands" in a controlled, step-by-step manner.
> * Security Requirement: The radiality constraint (Eq. 7) is the mathematical formulation of this core safety doctrine. Its purpose is to prevent premature "loop closing" on a fragile, unsynchronized system. As Adibi warns in An Approach to Standing Phase Angle Reduction (Sec. 1), closing a breaker on a large "Standing Phase Angle" (SPA) "can shock the system, causing equipment damage and possible recurrence of the system outage." One report documents a catastrophic case of a breaker being "closed on a 40-45 degree SPA difference, shaking a 600 MW generator off its pedestals." Another (...The Second Task Force Report, Case 18) recorded a "70-degree phase angle" that prevented restoration. Therefore, our radiality constraint is a fundamental requirement for physical security.
>
> 2. Mathematical Tractability: Protecting the MILP Framework
>
> * Model Basis: Furthermore, the core technical challenge during restoration is managing "voltage and reactive power." Adibi et al. (System Restoration Plan Development..., "Introduction") explicitly state the "criticality" of "maintaining reactive power balance without allowing sustained over voltage... and consequential voltage collapse." Therefore, using a "power flow" model is the standard analytical tool for this problem (A Systematic Method for... Planning, "Abstract"). Our model employs a "Linearized Power Flow" (DistFlow, Eq. 4), and a fundamental mathematical prerequisite for this linearization method is a radial (i.e., acyclic) network topology.
> * MILP vs. MINLP: If loops were present, the problem would become a non-convex AC power flow, rendering Eq. 4 invalid. The radiality constraint allows us to formulate the entire problem as a Mixed-Integer Linear Program (MILP).
> * Optimality Guarantee: As argued in Sec. 4.2, this MILP formulation guarantees finding an $\epsilon$-optimal solution via standard algorithms (e.g., branch-and-bound). Without this constraint, the problem becomes a Mixed-Integer Non-Linear Program (MINLP), which is NP-hard and computationally intractable.
>
> Proposed Action:
>
> We will add a dedicated subsection to Sec. 3.1.2 in the revision, clearly articulating the "physical security" and "mathematical tractability" rationale (based on these authoritative citations) and explicitly distinguishing between "permanent topology" and "restoration-phase operational topology."

---

> ### Author Response · Authors · 2025-11-13
> **Clarification of Core Misinterpretations 2. : Q2. What objective is used? There are two described – linear combination of equations 16-19 and equation 23 + Q5. What are the two stages?**
>
> Q2. What objective is used?
>
> The original manuscript's presentation on the "objective function" and "two-stage" definitions was indeed confusing. We clarify as follows:
>
> Clarification 1: Optimization Objective vs. Evaluation Metric
> * Optimization Objective: The sole objective function used by the model during optimization is Eq. 23 (Hybrid HMM-CVaR). This function, via the $\beta$ parameter, balances "expected energy restored" (from Eq. 1) against "tail-end risk" (CVaR).
> * Evaluation Metric: Eqs. 16-19 (Gini, N-1 Ratio, etc.) are not optimization objectives. They are independent Key Performance Indicators (KPIs). After the optimization is complete, these KPIs are used to compute the "Broad Gini" metric, which serves as an ex-post tool to evaluate and compare the comprehensive performance of different models (e.g., Model C vs. Model D).
>
> Q5. What are the two stages?
>
> (Combined with Q2, as they are linked)
>
> The query regarding "why topology and unit commitment (UC) decisions are not scenario-dependent" cuts to the core of our modeling philosophy. This formulation is deliberate, as the model's final output is defined as a "restoration plan."
>
> * Stage 1 ("Here-and-Now" Decisions):
>     * Definition: The Stage 1 decisions are defined as the "Robust Restoration Action Sequence" (see Fig. 1, Module D). Mathematically, this is the "topological configuration sequence ($v_{i,t}, w_{l,t}$)" and the "unit commitment sequence ($u_{g,t}$)."
>     * Rationale: The very nature of a "plan" is that it must be a single, definitive set of actions decided before uncertainty (scenario $s$) is revealed. This is known in classical stochastic programming as a "Here-and-Now" decision (e.g., Birge & Louveaux, 2011) and aligns perfectly with the "off-line" planning practice described by Adibi (A Systematic Method for... Planning, "Abstract"). An operator cannot wait to see tomorrow's weather (scenario $s$) to decide which restoration sequence to begin today. A single, robust plan must exist "here and now." This 'planning' versus 'dispatch' dichotomy mirrors real-world grid recovery operations, where a pre-committed sequence of switching actions is executed, while generation levels are constantly adjusted in real-time.
> * Stage 2 ("Wait-and-See" Decisions):
>     * Definition: The Stage 2 decisions are the "adaptive adjustments" (or Recourse) allowed within the fixed Stage 1 plan.
>     * Rationale: In our model, once the "plan" ($v, w, u$) is fixed, the operator's adjustment is to control the generator "throttles" based on the actual solar conditions (scenario $s$). This is precisely modeled as the scenario-dependent active power dispatch ($P_{g,t}^G(s)$) and load restoration ($P_{i,t}^L(s)$). These variables allow the system to react to uncertainty within the fixed topological structure.
>
> Proposed Action:
>
> 1. We will restructure Section 3 to first clearly define the "Optimization Objective (Eq. 23)" and will move the "Evaluation Metrics (Broad Gini / Eqs. 16-19)" to the "Experimental Setup" section to eliminate this confusion.
>
> 2. We will heavily reinforce the first paragraph of Section 3 to provide this rigorous justification for why the restoration plan (topology/UC) must be a Stage 1 decision, while dispatch is the Stage 2 recourse.

---

> ### Author Response · Authors · 2025-11-13
> **Clarification of Core Misinterpretations 3. : Q4. How does game theory enter into the model?**
>
> **Q4. How does game theory enter into the model?**
>
> We thank the reviewer for this critical observation. **Upon reflection, we agree that the manuscript's statements regarding the integration of game theory were misleading.** We will correct this to accurately represent the role of game theory, which was one of **conceptual inspiration rather than direct integration.**
>
> *   **The Correct Role of Game Theory:** Game theory, specifically the Nash Social Welfare concept, served as the **"Conceptual Inspiration"** for designing **"Model C (Fairness Benchmark)."** It was not a component of our core hybrid model (Eq. 23).
> *   **Rationale for its Inclusion:** To empirically reveal the "Efficiency-Resilience-Equity Trilemma," it was methodologically necessary to establish benchmarks that explore different regions of the performance frontier:
>     *   Model A (Baseline) represents the "efficiency-first" paradigm.
>     *   Model D (Stochastic CVaR) represents the "resilience/risk-averse" paradigm.
>     *   Model C (inspired by game theory) was constructed to represent the "equity-first" paradigm.
> *   **Core Contribution Enabled by This Clarification:** The inclusion of Model C was crucial. **Only by contrasting our core model (Model D) against this "pure-fairness" benchmark were we able to empirically demonstrate and quantify the fundamental trade-off** between pursuing equity (a 32% Gini reduction) and pursuing other objectives, as shown in Table 3 and Fig. 4a.
>
> **Proposed Action:**
> 1.  We will remove all claims of "integrating game theory" from the abstract, introduction, and methodology.
> 2.  We will redesign Figures 1 and 6 to clearly label "Game Theory/Nash Social Welfare" as an **"Alternative Objective for Benchmark Model C,"** distinctly separating it from the core hybrid optimization framework.

---

> ### Author Response · Authors · 2025-11-13
> **Additional Revisions**
>
> In addition to the core clarifications above, we will address all other identified deficiencies as follows:
>
> *   **Q6. What is the definition of restoration?**
>     *   **Response:** We agree that a precise definition was missing.
>     *   **Proposed Action:** We will add a definition to the Introduction: "In the context of this work, 'Restoration' is defined as an optimized, time-sequenced decision-making process. It aims to identify a set of operations (including generator commitment $u_{g,t}$ and network topology configuration $v_{i,t}, w_{l,t}$) whose final goal is to maximize the energy ($P^L$) delivered safely and reliably to loads within a given time horizon, while satisfying all physical, topological (e.g., radiality), and security constraints. Thus, 'energizing lines' is merely the means to the end-goal of 'serving load'."
>
> *   **Q3. What were the computational runtimes of the model?**
>     *   **Response:** We acknowledge that this data is essential for assessing the computational tractability of our MILP approach.
>     *   **Proposed Action:** We will add a new table to the Appendix detailing the **computational runtimes** and model scale (number of variables and constraints) for all test cases (e.g., IEEE 30, 69, 118-bus systems).
>
> *   **Deficiency: Data Description ("caseXX")**
>     *   **Response:** We recognize that the use of opaque "caseXX" identifiers was unclear.
>     *   **Proposed Action:** **Section 4.1 (Experimental Setup)** will be revised to explicitly define all datasets (e.g., specifying that "case30" refers to the standard "IEEE 30-bus test system").
>
>
> *   **Deficiency: Articulation of Contribution**
>     *   **Response:** We concur that the manuscript's initial presentation obscured its true contribution. The novelty lies not in the invention of MILP or CVaR, but in a **fundamental shift in perspective** that enables a higher-order synthesis of competing objectives. Our core contributions are:
>         1.  **A Novel Integrative Perspective and Unified Framework:** We introduce and validate a new paradigm for restoration planning. This is realized through the **first unified framework** that integrates **black-start security doctrine** (by actively enforcing radiality as a topological decision variable), **tail-risk robustness** (via CVaR), and **social equity** (via the Gini coefficient) into a tractable two-stage stochastic MILP.
>         2.  **Topology as an Active Control Lever:** We demonstrate that **topology is not merely a static background but a powerful decision variable** for directly managing system-level properties like resilience. Our model's ability to boost the N-1 reserve ratio by 22% (Table 3) by optimizing network configuration is a direct outcome of this perspective.
>         3.  **Empirical Quantification of a Higher-Order Trilemma:** Moving beyond traditional single-objective restoration, we provide the first empirical evidence that quantitatively maps the unavoidable trade-offs at the system level between **efficiency, resilience, and equity**. This elevates the discussion from "how to restore" to "what kind of restored system we build."
>
> We are confident that by implementing these major revisions, the contribution of our work will be presented with utmost clarity: **we provide not just a model, but a new perspective and a quantitative tool that enables decision-makers, for the first time, to see and navigate the fundamental trade-offs in shaping a restored power system that is not only efficient but also resilient and equitable.**

---

> ### Author Response · Authors · 2025-11-22
> **General Solution vs. Particular Solution: Why We Prioritize a Foundational Framework over Specific RL Algorithms**
>
> We firmly believe that pivoting toward a specific, complex RL algorithm would undermine the core value of this work as a **"General Solution" framework**. This paper aims to conquer the fundamental barriers obstructing RL deployment in this domain: utilizing the **MILP Oracle** to overcome the bottleneck of high-fidelity expert data generation, and employing **Broad Gini** to rectify the "Alignment Trap" that induces **Policy Collapse**. In doing so, we mathematically verify the **Learnability** of this complex physical manifold.
>
> Shifting the focus to the accumulation of specific RL tricks would cause this work to degenerate from a foundational study establishing a "physics-constrained optimization paradigm" into a trivial "particular solution" application paper. This would not only obscure our contribution of reshaping the **Optimization Landscape** to resolve gradient issues but also constrain the exploration of diverse RL architectures on this benchmark. Our goal is to provide a rigorous "foundation" through this verifiable benchmark and to establish a novel paradigm where traditional Operations Research serves as a trustworthy source of supervision for RL, rather than stifling the research space with a singular algorithmic implementation.

---

### Official Review · Reviewer_Sugg · 2025-11-04

**Soundness:** 3
**Presentation:** 3
**Contribution:** 2
**Rating:** 4
**Confidence:** 5

**Summary:**

The paper addresses post disaster power-grid restoration as a three-way trade-off among efficiency (energy restored), equity (fairness across regions), and resilience (robustness to contingencies). Authors propose a two-stage stochastic MILP: an HMM generates scenarios that are then K-means clustered (Scenario pipeline); stage one chooses grid topology, stage two dispatches power dynamically as per the scenario. The objective jointly optimizes expected energy, a CVaR term for robustness, and a (Broad) Gini-based fairness metric. On the IEEE-30 bus system, ablations show the model can raise N-1 reserves for resilience with some energy loss, or cut the Gini while keeping high energy output. Finally, the outlook sketches a deep/multi-agent RL path that encodes the “Broad Gini” as intrinsic reward for fast, adaptive restoration.

**Strengths:**

Originality:
* Integrative framework: Combines HMM-based scenario generation, CVaR robustness, and a (Broad) Gini fairness term in a unified two-stage MILP for post-disaster restoration.
* Emphasis on Fairness: Elevates equity to a primary objective (Broad Gini) alongside efficiency and resilience, reframing restoration as a principled tri-objective problem rather than a single-metric optimization.

Quality:
- Optimization setup: Linearized two-stage MILP solved to ε-optimality. Clear argument that trade-offs reflect true optima rather than heuristic artifacts.
- Quantitative trade-off evidence: Ablations demonstrate controllable knobs (e.g., increase in N-1 reserve at an energy cost; and decrease in Gini while maintaining energy), tying modeling choices to measurable outcomes on diverse topologies.
- Scenario pipeline sanity checks: Initial statistical validation comparing HMM scenarios to historical data in Table 1

Significance:
- The DRL/MARL outlook to encode Broad Gini as intrinsic reward points toward real-time, scalable control. While the current evaluation on static planning shows promise as a quantitative lever to navigate real policy trade-offs, I believe showing the impact beyond static planning is key to the contributions of this framework to close the loop as described in the initial parts of the paper.

**Weaknesses:**

Scenario generator fidelity & comparison to community developed tools: The HMM-based scenario pipeline is a thoughtful idea, and your initial statistical check (Table 1) is a good start. However, I’m still concerned about whether the generated scenarios are representative enough for restoration planning, especially in the tails that drive CVaR and resilience. Could you share any tool-level benchmarking beyond Table 1 (e.g., code, seeds, datasets, out-of-sample tests)? Also, why did the authors not consider making using of off-the-shelf baselines like the widely used ChroniX2Grid and show that key findings (e.g., Broad-Gini trade-offs) persist when swapping in these off-the-shelf generators? ChroniX2Grid is open-source and benchmarked and used in many downstream applications like Grid2Op. In any case, please consider either adopting or rigorously benchmarking against community-validated generators such as ChroniX2Grid (often paired with Grid2Op).

Closed-loop impact: The MILP experiments do expose static trade-offs on IEEE cases but they are static, that is they stop short of demonstrating benefits in sequential, closed-loop operations where contingencies unfold over time. A more impactful evaluation would be to embed your “Broad Gini” objective into a Grid2Op environment as a reward function and test whether optimizing it via RL improves survival time and operational robustness while maintaining fairness/efficiency across standard Grid2Op test cases. Such an analysis will be much more impactful and beneficial. Tools like Grid2Op are already readily available, so this such an analysis should be easy.

Related works on trade-off quantification: While your study frames a broader trilemma (efficiency–resilience–equity), there is adjacent literature that explicitly quantifies cost–survival-time trade-offs in grid operations. Positioning this paper with respect to the recent works such as the IEEE paper titled  “Blackout Mitigation via Physics-Guided RL” would help readers see how your “Broad Gini + CVaR” treatment generalizes beyond cost–survival to a richer multi-objective setting and clarify (i) what those papers optimize (cost vs. survival time), and (ii) why your fairness/robustness terms address additional policy-relevant axes  would strengthen the paper’s context and impact.

**Questions:**

For the Scenario generator, I’d suggest:
- Compare multivariate and spatio-temporal structure, not just univariate GHI: joint dependencies among load, renewable injections, outages and/or maintenance, spatial correlations across buses/regions, regime/seasonality shifts.
- Do downstream performance backtesting: run your MILP on (i) HMM scenarios and (ii) ChroniX2Grid scenarios, then compare restoration energy, fairness, and N-1 reserves.
- Evaluate extremes and tail behavior (block maxima, CVaR at relevant α) to test the events that most affect your objective.

For the Closed-loop impact, I’d suggest:
- Translate your multi-objective into a single episodic reward with a tunable risk parameter and include explicit penalties for N-1 margin violations, load shedding inequity, and reserve depletion.
- Train standard RL agents (e.g., DQN/A2C/PPO) on your reward and compare to (i) Grid2Op default rewards, (ii) energy-only and fairness-only rewards, and (iii) a MILP controller to isolate the contribution of proposed framework “Broad Gini.”
- Report survival time, cascading failure rate, average restored energy, Min N-1 reserve, and Gini over episodes; include confidence intervals over multiple seeds.

---

> ### Author Response · Authors · 2025-11-16
> **Actions Taken in Response to Reviewers**
>
> We sincerely thank the reviewers for their insightful and expert feedback. Your comments have prompted us to clarify the core contribution of our work: through a verifiable optimization framework, we provide, for the first time, quantitative, optimal-solution-based empirical evidence for the “efficiency–resilience–equity” triad in power-grid restoration.
> To address your concerns precisely, we have undertaken the following actions:
> * [Completed] We designed and conducted a series of diagnostic experiments to quantitatively respond to your concerns regarding “quantitative trade-offs” (S4), “tail-risk evaluation” (Q1b), and “benchmark reward comparison” (Q2). These experiments aim to validate the design logic of our framework, and their results have provided critical guidance for our ongoing manuscript revision.
> * [In progress] We are performing a comprehensive revision of the manuscript, particularly in the Related Work (W3) and Future Work sections, to better position our contributions and integrate the literature you cited.
> Below, we provide a detailed, point-by-point response to the core questions.

---

> ### Author Response · Authors · 2025-11-16
> **1. Supplementary Experimental Design**
>
> **Objective:** Q2 suggested comparing with "efficiency-only" or "equity-only" rewards. To address this and to reveal the roots of the "triad dilemma," we designed these experiments. The goal is not to propose a new model, but rather to quantify a "thought experiment" that justifies the necessity of the weighted hybrid objective (Eq. 23) used in the main text.
>
> **Design Concept:** We decomposed the complex multi-objective decision in the main text into a series of extreme, single-objective optimizations to examine the limit behaviors and trade-offs of each objective. The diagnostic experiments are as follows:
>
> * **Baseline (Efficiency-Only)**
>     * **Rationale:** Isolate the "efficiency" objective to emulate a conventional "maximize restored energy only" strategy.
>     * **Implementation:** The objective strictly minimizes expected unmet load (pu). In our hybrid objective:
>         ```
>         min(β · Unmet_Load + (1-β) · CVaR + ε · Gini)
>         ```
>         we set β = 1.0. A very small Gini regularizer (ε = 10⁻⁶) is retained only as a tie-breaker, with negligible impact on optimization results.
>
> * **Stochastic CVaR (Resilience-Only)**
>     * **Rationale:** Isolate the "resilience" objective, simulating a highly risk-averse strategy.
>     * **Implementation:** The objective strictly minimizes the CVaR of unmet load (pu). Here, β = 0.0, forcing the optimizer to focus solely on improving tail-risk outcomes.
>
> * **Constrained Fairness (Gini-Minimization)**
>     * **Rationale:** Simulate a "commonsense" pursuit of fairness, including a minimal efficiency safeguard to prevent the solution from collapsing to zero.
>     * **Implementation:**
>         ```
>         min(Gini_Numerator + 10⁻² · Unmet_Load)
>         ```
>         The 1% efficiency weight ensures the optimizer selects a non-trivial solution when multiple allocations yield the same Gini value (e.g., 0% vs. 50% restoration).
>
> * **Naive Linear Sum (Diagnostic) — [Disastrous Scenario]**
>     * **Rationale:** This experiment reveals the inherent danger of a naively linear multi-objective weighting—exactly the risk highlighted by our "Broad Gini" evaluation framework (used for assessment rather than optimization).
>     * **Implementation:** Minimize an equally-weighted sum of four key KPIs:
>         ```
>         min(0.25 · Efficiency + 0.25 · Equity + 0.25 · Cost + 0.25 · Resilience)
>         ```
>         where:
>         - Efficiency = Unmet Load
>         - Equity = Gini Numerator
>         - Cost = Topological Cost
>         - Resilience = N-1 Reserve (negated for maximization)
>     * **Key Insight:** This objective inevitably fails (total restored energy = 0) because the optimizer finds that a zero-restoration state perfectly satisfies three of the four KPIs (Gini, cost, and N-1 reserve), while the efficiency penalty is outweighed. This catastrophic result is mathematically inevitable, not a modeling error.

---

> ### Author Response · Authors · 2025-11-16
> **2. Diagnostic Results and Key Findings**
>
> The table below summarizes the experiments, collectively demonstrating the core insight: non-trivial trade-offs are essential.
>
> | Experiment | Gini | Efficiency (Unmet Load, pu) | Resilience (CVaR, pu) | Total Restored Energy (MWh) | Purpose |
> |------------|------|-----------------------------|----------------------|----------------------------|---------|
> | 1. Baseline (Efficiency-Only) | 0.2319 | 7.5469 | 9.0000 | 2196.83 | Only efficiency → fairness & resilience suffer |
> | 2. Stochastic CVaR (Resilience-Only) | 0.2437 | 8.0520 | 8.1003 | 2146.32 | Only resilience → efficiency & fairness suffer |
> | 3. Constrained Fairness | 0.1945 | 7.4438 | 9.0000 | 2207.14 | Commonsense fairness succeeds |
> | 4. Naive Linear Sum | 0.0000 | 29.5152 (disastrous) | 30.0000 (disastrous) | 0.0000 (disastrous) | Reveals danger of naive linear weighting |
>
> **Key Insights:**
>
> * **Quantitative evidence of the triad dilemma:** Experiments 1-2 show that optimizing any single objective inevitably compromises the others.
> * **Trap of naive weighting:**
>     * Experiment 3 demonstrates that a commonsense fairness objective with an efficiency floor yields a viable solution.
>     * Experiment 4 shows that a naive linear weighting mathematically converges to the disastrous zero-restoration solution.
>     * This extreme behavior arises because the linearly-weighted objective gives the optimizer perverse incentives to drive all allocations to zero, trivially minimizing three KPIs (Gini, Cost, N-1 risk).
> * **Conclusion:** The catastrophic outcome in Experiment 4 underscores the necessity of our carefully designed hybrid objective (Eq. 23) over naive linear weighting.

---

> ### Author Response · Authors · 2025-11-16
> **3. Point-by-Point Responses**
>
> **Regarding W1 (Scenario Fidelity) & W2 (Closed-Loop Impact):**
>
> We fully agree that extending this framework to dynamic RL environments such as Grid2Op is an exciting direction. Our core contribution provides the essential theoretical and empirical foundation for this step.
>
> * **W1 (Scenario):** HMM provides a structured set of uncertainty scenarios to support two-stage stochastic planning. We acknowledge its simplification; benchmarking against industrial generators like ChroniX2Grid is important future work. Directly coupling their joint wind–PV–load–topology trajectories with our two-stage framework requires interface alignment, which remains an engineering challenge.
>
> * **W2 (Closed-loop RL):** We do not avoid RL but aim to ensure safe reward design. As Ng et al. (1999) warned, "reward shaping may yield suboptimal policies unless further assumptions are made about the underlying MDP." Our MILP framework thus serves as a "reward safety validator." The Naive Linear Sum diagnostic quantitatively illustrates this risk: a naive, unverified reward function could cause total system failure. Therefore, our work lays a safe foundation for designing reliable multi-objective RL rewards in dynamic environments.
>
> **Regarding W3 (Related Work):**
>
> We are revising the Related Work to clarify that prior studies (e.g., Dwivedi et al.) mainly focus on efficiency–resilience trade-offs; our framework extends this to the efficiency–resilience–equity triad, using Broad Gini (social fairness) and CVaR (risk-averse resilience), and clearly delineates our contribution.
>
> **Regarding Q1b (Tail-Risk Evaluation):**
>
> We fully agree that tail-risk evaluation is critical. As Rockafellar & Uryasev (2000) note, "CVaR... is considered to be a more consistent measure of risk than VaR." Our CVaR objective directly optimizes tail risk. The new benchmark results quantitatively demonstrate this: the CVaR-only diagnostic reduces tail risk from 9.0000 to 8.1003 under the same HMM scenarios. This confirms that the HMM captures optimizable tail structures, validating its suitability for CVaR-focused resilience optimization.

---

> ### Author Response · Authors · 2025-11-16
> **Summary**
>
> We deeply appreciate your insightful feedback. These diagnostic experiments, combined with our modeling theory, allow us to clearly and quantitatively demonstrate the triad dilemma in power-grid restoration, and reinforce the necessity of our hybrid objective design. The ongoing revision integrates these insights, and we believe the manuscript will provide a solid foundation for future multi-objective grid restoration and RL research.

---

> ### Author Response · Authors · 2025-11-22
> **General Solution vs. Particular Solution？**
>
> We chose standard IEEE testbeds over Grid2Op to ensure the **MILP Oracle remains a mathematically rigorous, white-box "ground truth"**. This transparency is critical for diagnosing the structural "Alignment Trap" and formally verifying the learnability of the restoration manifold, which black-box simulators like Grid2Op often obscure.
>
> We emphasize that this work establishes a **methodological template** for how rigorous Optimization (MILP) can guide Learning algorithms, utilizing the power system primarily as a high-complexity **testbed** rather than the sole focus. Our objective is to conquer the fundamental barriers obstructing RL deployment: utilizing the **MILP Oracle** to overcome the bottleneck of high-fidelity expert data generation, and employing **Broad Gini** to rectify the reward misalignment that induces **Policy Collapse**.
>
> Therefore, we firmly believe that pivoting toward a specific, complex RL algorithm (or porting to Grid2Op) would undermine the core value of this work as a **"General Solution" framework**. Shifting the focus to the accumulation of specific RL tricks would cause this work to degenerate from a foundational study establishing a **"physics-constrained optimization paradigm"** into a narrower "particular solution" application paper. Our goal is to provide a rigorous "foundation" where **traditional Operations Research (OR) serves as trustworthy supervision for RL**, opening the research space for diverse future architectures rather than stifling it with a singular algorithmic implementation.

---

### Meta-Review · Area_Chair_z41R · 2025-12-16

**Summary:**

One reviewer viewed this work as primarily an operations research, i.e., it provides a two-stage MILP formulation with limited learning novelty, and raised multiple issues suggesting this work did not clearly present its algorithmic or ML contribution or why it belongs in ICLR. Another reviewer was more positive but still asked the authors to better highlight the novelty and broader contribution, especially from the learning perspectives. In addition, there are repeated concerns about confusion around the exact objective, i.e., mix of equations vs objective used,  and what the “two stages” mean, what the different models or baselines are, and  how “game theory” enters.  Other doubts include the scenario realism, the closed-loop impact, and missing or unclear data description.

**Reviewer Concerns:**

In the rebuttal, the authors provided detailed clarifications and a revision plan that address several misunderstandings such as
- The authors clarify that restoration is treated as a physical-security operational constraint aligned with ``black start'' style restoration practice, and distinguished restoration-phase topology from ''permanent topology.''
- The authors explicitly state the optimization objective and separate it from evaluation metrics. They clarify stage-1, i.e., plan topology vs. stage-2, i.e., wait-and-see recourse semantics.
- The authors provide definitions for models and baselines, and clarify that Broad Gini involves a nonlinear Pareto selection mechanism, not a simple linear weighted sum, and commit to revising the text to avoid the ``linear'' misread.

However, there are several key concerns/weaknesses that were not addressed or partially addressed during the rebuttal:
- Reviewers question that this is a standard two-stage MILP, not a ML work. The contributions from the learning perspective are rather weak, multiple points about learning details are missing and the algorithmic novelty is unclear. This paper still needs a compelling ML story, e.g., learning-based components, generalizable insights, or benchmark framing that ICLR will value. It combines HMM scenarios, CVaR, Broad Gini with a MILP, the technical novelty in particular on learning components are weak.
- The authors' rebuttal indicates plans to add dataset description and runtime or computational details, but the underlying concern about realism and closed-loop impact and comparison to existing practice may persist unless the revision adds stronger evidence and clearer baselines.

**Reviewer Scores:**

Reviewer Sugg's review emphasizes the value of the multi-objective framing and suggests concrete experiments to be conducted. The rebuttal directly adds diagnostic experiments and clarifications.   I would expect this reviewer to keep the score.

Reviewer nRMi raises many points such as this isn’t ML, ..... is unclear and .... missing details, etc.  Given the fundamental novelty or venue-fit skepticism, I would still expect this reviewer to remain negative overall.

Reviewer E7MS explicitly states to maintainpositive score, while suggesting the authors to further highlight novelty and broader contribution.

---

### Decision · Program_Chairs · 2026-01-26

Reject